# Nitric oxide orchestrates metabolic rewiring in M1 macrophages by targeting aconitase 2 and pyruvate dehydrogenase

Erika M. Palmieri[1], Marieli Gonzalez-Cotto[1], Walter A. Baseler[1], Luke C. Davies[1,2], Bart Ghesquière[3,4], Nunziata Maio[5], Christopher M. Rice[1,6], Tracey A. Rouault [5], Teresa Cassel[7], Richard M. Higashi[7], Andrew N. Lane [7], Teresa W.-M. Fan [7], David A. Wink[8] & Daniel W. McVicar [1]*

Profound metabolic changes are characteristic of macrophages during classical activation and have been implicated in this phenotype. Here we demonstrate that nitric oxide (NO) produced by murine macrophages is responsible for TCA cycle alterations and citrate accumulation associated with polarization. $^{13}$C tracing and mitochondrial respiration experiments map NO-mediated suppression of metabolism to mitochondrial aconitase (ACO2). Moreover, we find that inflammatory macrophages reroute pyruvate away from pyruvate dehydrogenase (PDH) in an NO-dependent and hypoxia-inducible factor 1α (Hif1α)-independent manner, thereby promoting glutamine-based anaplerosis. Ultimately, NO accumulation leads to suppression and loss of mitochondrial electron transport chain (ETC) complexes. Our data reveal that macrophages metabolic rewiring, in vitro and in vivo, is dependent on NO targeting specific pathways, resulting in reduced production of inflammatory mediators. Our findings require modification to current models of macrophage biology and demonstrate that reprogramming of metabolism should be considered a result rather than a mediator of inflammatory polarization.

[1] Leukocyte Signaling Section, Cancer & Inflammation Program, National Cancer Institute, Frederick, MD, USA. [2] Division of Infection & Immunity, School of Medicine, Cardiff University, Tenovus Building, Heath Park, Cardiff CF14 4XN, UK. [3] Metabolomics Expertise Center, Vesalius Research Center, VIB, 3000 Leuven, Belgium. [4] Metabolomics Expertise Center, Department of Oncology, KU Leuven, 3000 Leuven, Belgium. [5] Molecular Medicine Branch, Eunice Kennedy Shriver National Institute of Child Health and Human Development, Bethesda, MD, USA. [6] School of Cellular and Molecular Medicine, Faculty of Life Sciences, University of Bristol, Bristol BS8 1TD, UK. [7] Department of Toxicology and Cancer Biology and Markey Cancer Center and Center for Environmental and Systems Biochemistry, University of Kentucky, Lexington, KY, USA. [8] Chemical and Molecular Inflammation Section, Cancer and Inflammation Program, National Cancer Institute, Frederick, MD, USA. *email: mcvicard@mail.nih.gov

Macrophages are important players in the regulation of immune responses and tissue homeostasis. Production of proinflammatory mediators, microbicidal activity, and release of reactive oxygen species (ROS) have been associated with "classical" activation while "alternative" activation mediates healing, humoral responses and tumor progression[1]. Generally, classical "M1" macrophages utilize glycolysis, whereas "M2", differentiated in interleukin 4 (IL4), rely on oxidative phosphorylation (OXPHOS)[2]. Shifts toward Warburg metabolism have been well characterized in lipopolysaccharide (LPS) or LPS and interferon (IFN)γ-activated macrophages[3,4]. The concomitant increases in glucose consumption and lactate production, together with reduced OXPHOS have been termed "commitment" to a glycolytic state, crucial for supporting many aspects of macrophage polarization[3,5,6]. Nonetheless studies have reported complex metabolic footprints in macrophages activated under certain stimuli and tissue environments[7,8]. As a result, the possibility of modulating metabolic pathways to skew immune cells towards anti-tumoral phenotypes has driven interest in both immunology and cancer biology.

Macrophage reprogramming is associated with accumulation of key metabolites tied to macrophage functions[2,9]. The accumulation of citrate, succinate and itaconate are considered result of dysfunctional or "broken" citric acid cycle, also known as TCA cycle, as defined by reduction in carbon transition from citrate to alpha-ketoglutarate (α-KG), a decline in isocitrate dehydrogenase (IDH1) messenger RNA (mRNA) levels, and decreased succinate dehydrogenase (SDH) activity[10]. Accumulated citrate supports synthesis of lipids for production of inflammatory mediators[11,12] while succinate has been implicated in the stabilization of HIF1α[4], which in turns promotes production of IL1β and inhibits pyruvate dehydrogenase (PDH)[13] limiting acetyl coenzyme A for the TCA cycle. Itaconate, the product of decarboxylation of cis-aconitate by IRG1[14,15], possesses antimicrobial activity[16], inhibits SDH, thereby promoting succinate accumulation and regulating ROS production[14,17,18] and acts as an immunomodulatory metabolite[19,20]. Despite the many functions ascribed to metabolite changes during polarization, the underlying mechanisms of reprogramming are still poorly defined.

More than 30 years ago, NO-mediated inhibition of complex I (CI), II, III, and IV of the mitochondrial ETC[21–24] was described as a mechanism by which activated macrophages kill tumor cells[21]. In M1 macrophages NO is largely the product of inducible nitric oxide synthase (iNOS or Nos2), which, unlike neuronal NOS (nNOS or Nos3) or endothelial NOS (eNOS or Nos1), strongly increases during stimulation with LPS[25]. We previously demonstrated that NOS-derived NO is both necessary and sufficient for repression of OXPHOS during M1 polarization in bone marrow-derived macrophages (BMDMs). Moreover, we found that IL10 tunes glycolytic commitment by limiting the production of NO[3]. Given the ability of NO to regulate mitochondrial respiration, we investigated the possibility that this soluble gas might play a role in other aspects of macrophage programming. We find that, upon stimulation with LPS or LPS/IFNγ, NO production dictates intracellular routes of carbon utilization skewing mitochondrial metabolism at aconitase 2 and potently suppressing carbon flow into the TCA via PDH in a HIF1α-independent manner. Our work reveals that NO orchestrates the accumulation of several intermediates, ultimately leading to suppression and loss of mitochondrial ETC complexes. Moreover, we find that cytokine production is unaffected in macrophages that fail to rewire their mitochondria demonstrating that loss of OXPHOS and broken TCA cycle are results of inflammatory polarization rather than mediators of the process.

## Results

**Nos2$^{-/-}$ macrophages show intact metabolism and inflammatory machinery.** In order to assess what aspects of metabolic programming other than suppression of OXPHOS might be driven by NO we carried out detailed metabolic analysis. Steady-state metabolomics contrasting WT and Nos2$^{-/-}$ BMDMs stimulated with LPS for 24 h demonstrated profound differences. Analysis of metabolites involved in arginine metabolism showed that BMDMs accumulate citrulline as result of conversion of arginine during the production of NO by NOS2 (Fig. 1a). Macrophages lacking NOS2 had low citrulline and increased ornithine, consistent with alternative destination for arginine through arginase, whereas putrescine levels increased to the same extent in WT and Nos2$^{-/-}$ (Fig. 1a). Consistent with previous reports, activated WT were glycolytic (Fig. 1b) and Nos2$^{-/-}$ macrophages had higher levels of glycolytic intermediates, but showed rates of glycolysis comparable to WT (Fig. 1b and Supplementary Fig. 1A–D) with greater glycolytic reserve (Supplementary Fig. 1E). Quantitative PCR (qPCR) analysis showed that changes in metabolites correlated with upregulation of glycolytic genes in both WT and Nos2$^{-/-}$ (Supplementary Fig. 1F). Citrate, cis-aconitate, succinate, and itaconate accumulated in LPS-treated WT, while α-KG declined (Fig. 1c), the latter indicator of the reported "break" in the TCA cycle[26]. In Nos2$^{-/-}$, itaconate levels were higher than WT, and succinate accumulated whereas citrate did not (Fig. 1c) and the decline in α-KG was absent.

Reduced α-KG is indicative of the TCA break, but not definitive proof. Therefore, we performed stable isotopic labeling using uniformly labeled $^{13}$C-glucose ([U-$^{13}$C]) in untreated and LPS/IFNγ-stimulated WT and Nos2$^{-/-}$ BMDMs. Taking into account the total amount of citrate synthesized from glucose as pyruvate-derived acetyl-CoA enters the TCA ($m + 2$ isotopologues, Fig. 1d), WT macrophages showed little cis-aconitate and almost no α-KG derived from $^{13}$C citrate, confirming TCA break (Fig. 1e). Importantly, activated Nos2$^{-/-}$ did not exhibit any apparent break, as proportions of $^{13}$C-glucose-derived α-KG were similar to that of unstimulated cells. This effect was most evident when expressed as ratios of $^{13}$C citrate to α-KG (Fig. 1f). As a result, $m + 2$ glutamate, succinate, fumarate, malate and $m + 1$ itaconate were barely detectable in stimulated WT (Fig. 1g). Taken together, our results indicate intact glycolytic and TCA machinery in Nos2$^{-/-}$ macrophages, regulation of itaconate production by NOS2, and a role for NO in citrate accumulation during inflammatory macrophage polarization.

Metabolic changes have been suggested to be critical in the development of inflammatory macrophages[2]. As we find that Nos2$^{-/-}$ fail to undergo large scale mitochondrial metabolic rewiring, we expected their ability to differentiate into inflammatory macrophages to be impaired. Surprisingly, transcriptional profiling of stimulated Nos2$^{-/-}$ macrophages showed upregulated genes (Fig. 1h) enriched in pathways related to cytokine production and establishment and maintenance of the inflammatory response (Table 1 and Supplementary Table 1) while regulation of M2-associated genes[27–29] was unaffected by the absence of NO (Supplementary Fig. 1G). Assessment of secreted inflammatory mediators confirmed enhanced inflammatory state of Nos2$^{-/-}$ macrophages, including increased production of IL1β, IL6, IL12p40, macrophage inflammatory protein-α (MIP1α/CCL3) and monocyte chemoattractant protein-1 (MCP1/CCL2). Tumor necrosis factor (TNFα), IL10, and chemokine C-X-C motif ligand-1 (KC/CXCL-1) production was unaffected (Fig. 1i).

In addition, gene ontology enrichment analysis showed lower expression of nuclear factor erythroid 2-related factor 2 (Nrf2) target genes in absence of NOS2 (Table 1), suggesting that while

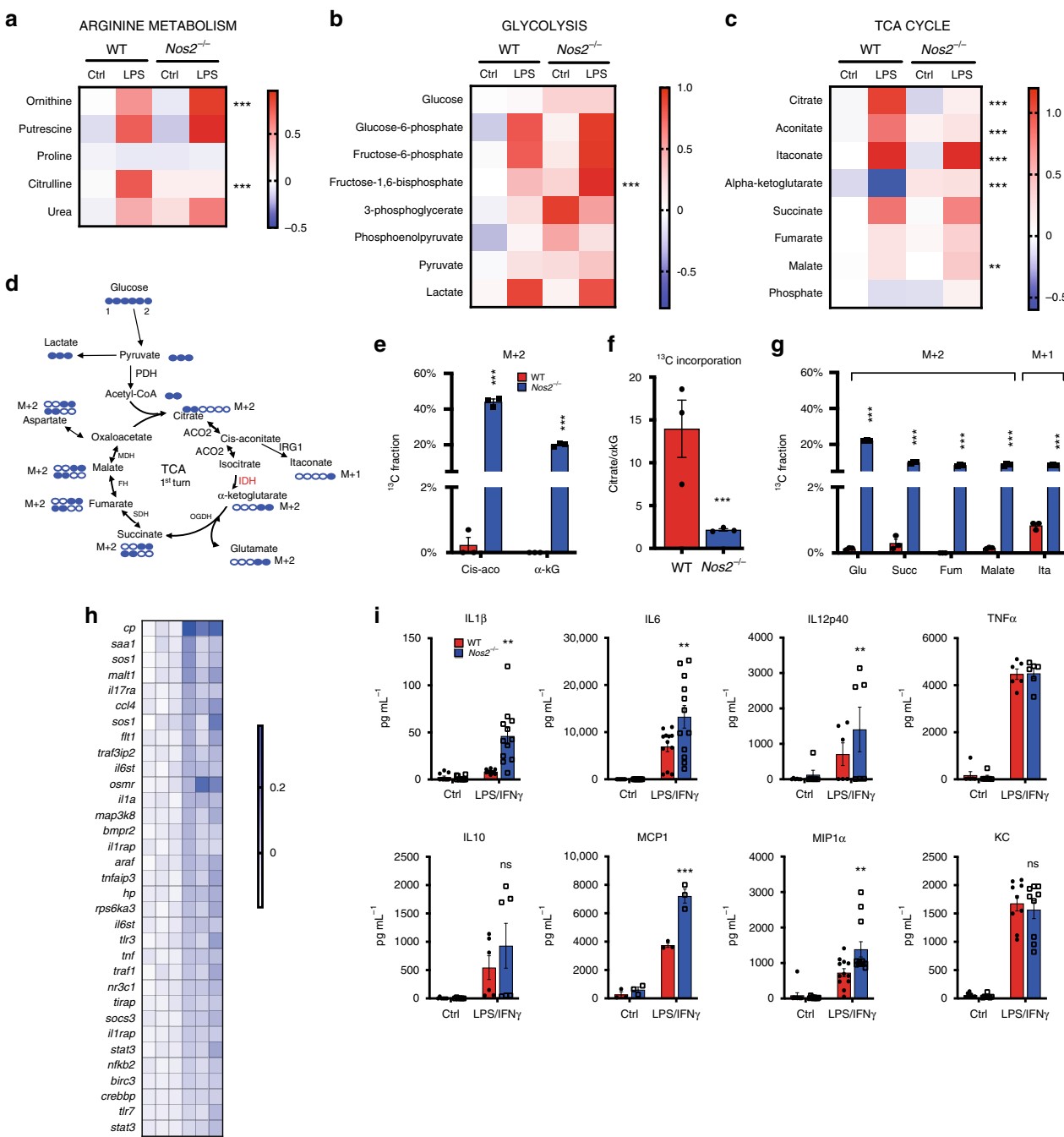

**Fig. 1 Nos2−/− macrophages show intact metabolism and inflammatory machinery.** Heat-maps of the log10 ratio from the average peak areas from Gas Chromatography-Mass Spectrometry (GC-MS) analysis of metabolites associated with the arginine metabolism (**a**), glycolysis (**b**), and citric acid cycle (**c**) from bone marrow-derived macrophages (BMDMs) from wild-type (WT) and Nos2−/− mice activated with LPS for 24 h compared to unstimulated (ctrl). **d** Schematic illustration of atom transitions in central metabolism using uniformly labeled 13C-glucose ([U-13C]) (labeled carbons are indicated in blue) as tracer for determination of mass isotopologue distributions (MID) to infer relative intracellular fluxes through oxidation of pyruvate. PDH pyruvate dehydrogenase, ACO2 aconitase 2, IDH isocitrate dehydrogenase, OGDH oxoglutarate dehydrogenase, SDH succinate dehydrogenase, FH fumarate hydratase, MDH malate dehydrogenase. **e–g** WT and Nos2−/− BMDMs were activated with LPS + IFNγ and cultured with labeled tracer. Bars show evaluation of the [U-13C] glucose-derived carbon incorporation (percentage) into m + 2 isotopologues of the indicated TCA intermediates. **e** Percentage of 13C in m + 2 isotopologues of cis-aconitate (cis-aco) and α-ketoglutarate (α-kG). **f** Bar graph depicting the ratio between citrate and α-kG for 13C fraction of the total metabolite level. **g** m + 2 isotopologues of glutamate (glu), succinate (succ), fumarate, malate and in m + 1 itaconate (ita) in activated WT and Nos2−/− BMDMs. Data in **a–c** (n = 5 biological replicates per group) and **e–g** (n = 6) were analyzed by two-way ANOVA (interaction < 0.0001) (Sidak's post-tests). **h** Expression profile of selected inflammatory genes from microarray analysis (see Table1) shown as heatmap of log2 fold changes of LPS + IFNγ-stimulated Nos2−/− BMDMs compared to WT (n = 3). **I** The concentration of IL1β, IL6, IL12p40, TNFα, IL10, MCP1, MIP1α, and KC secreted into the culture media when BMDMs were stimulated for 24 h. Data are pooled from three experiments (n = 7–8) and were analyzed by two-way ANOVA with Sidak's post-tests. All error bars display mean ± SEM. Source data are provided as a Source Data file.

**Table 1 Enriched canonical pathways of differentially expressed genes.**

| Canonical pathway | z-score | p-value | Associated gene number |
|---|---|---|---|
| NF-κB signaling | 3.207 | 2.40E-05 | 14 |
| Acute phase response signaling | 2.887 | 2.24E-05 | 14 |
| Role of IL17F in allergic inflammatory airway diseases | 2.449 | 4.07E-05 | 7 |
| Toll-like receptor signaling | 1.342 | 3.09E-05 | 9 |
| TNFR2 signaling | 1.000 | 3.09E-05 | 6 |
| NRF2-mediated oxidative stress response | −1.069 | 7.41E-06 | 32 |

Ingenuity pathway analysis (IPA) of the score for the likelihood that genes belonging to a specific canonical pathway category were different after 8 h of LPS + IFNγ stimulation of $Nos2^{-/-}$ and WT BMDMs. Positive or negative z-score value indicates that a function is predicted to be increased or decreased in $Nos2^{-/-}$ relative to WT activated cells. The significance of canonical pathways was determined by IPA's default threshold [−log (p-value) > 1.3] (p < 0.05). The associated gene number represents the number of differentially expressed genes involved in the respective canonical pathway.

polarization of WT macrophages elicits an anti-oxidant response, stimulation of $Nos2^{-/-}$ cells does not.

**The TCA Break is due to NO targeting of mitochondrial aconitase**. Previous reports have suggested that the TCA break is due to downregulation of Idh1[26]. Having determined that the break is NOS2 dependent, we investigated possible mechanisms for this process. Consistent with other studies, we detected a decline in *Idh1* mRNA in stimulated macrophages (Fig. 2a), but in contrast to the TCA break, this was NOS2 and NO-independent (Fig. 2a and Supplementary Fig. 2A, B). Accordingly, the iNOS inhibitor Aminoguanidine (AG) had no effect on IDH enzymatic activities (Supplementary Fig. 2C).

In contrast to IDH1, mitochondrial aconitase (ACO2) is a known target of NO[22,24]. Therefore, we tested the possibility that NO-mediated suppression of ACO2, rather than IDH1, might be responsible for the TCA break. Although aconitase mRNA levels were not affected by LPS/IFNγ (Supplementary Fig. 2D), WT M1 BMDMs had reduced protein levels of ACO2 (Fig. 2b). Moreover, LPS/IFNγ or direct application of the NO donor DETA/NO to resting cells substantially reduced ACO2 enzymatic activity without affecting cytosolic ACO1 (Fig. 2c–e). ACO2 activity remained unchanged in stimulated $Nos2^{-/-}$ (Fig. 2c) and in WT cells in the presence of AG (Fig. 2e). Finally, addition of NO donor to cultures of stimulated $Nos2^{-/-}$ macrophages was sufficient to decrease ACO2 activity in concentration dependent manner (Supplementary Fig. 2E). Consistent with NO-suppressing ACO2 through disruption of its iron-sulfur cluster [Fe₄-S₄][24,30], treatment with ferrous iron ($Fe^{2+}$) and reducing agents reconstituted ACO2 (Supplementary Fig. 2F). To further confirm the break at ACO2, we used direct fueling of permeabilized macrophages. Supplying stimulated WT cells with citrate (state 3 respiration) diminished respiration as compared to resting macrophages (Fig. 2f and Supplementary Fig. 2G). However, addition of isocitrate increased OCR, indicating that the TCA break is upstream of isocitrate. In addition, stimulated $Nos2^{-/-}$, or WT macrophages treated with AG, maintained the ability to respire citrate (Fig. 2f, g and Supplementary Fig. 2G). To prove the impact of ACO2 inhibition on the TCA, we stimulated $Nos2^{-/-}$ in the presence of the aconitase inhibitor fluoroacetate (FA)[31] (Supplementary Fig. 2H). After confirming intracellular delivery of FA (Supplementary Fig. 2I), we observed FA-dependent citrate and cis-aconitate accumulation together with decreased α−KG (Fig. 2h). Moreover, pharmacologic inhibition of ACO was sufficient to decrease respiration (Fig. 2i and Supplementary Fig. 2J). Interestingly, the magnitude of these changes only partially mirrors the effect of DETA/NO as ACO blockade alone did not affect itaconate or succinate and was not sufficient to decrease downstream TCA intermediates (Fig. 2h and Supplementary Fig. 2K).

Altogether these data unequivocally show that ACO2 rather than IDH1 is the TCA breakpoint in M1 macrophages, that this is responsible for increases in citrate and reductions in α−KG, and that NO is necessary and sufficient to mediate these effects.

Moreover, detection of rotenone sensitive, isocitrate-mediated respiration (Fig. 2f, g) indicated that CI, CIII, and CIV of the ETC were still functional even after over-night stimulation.

**Absence of NO promotes pyruvate oxidation via PDH**. To investigate potential roles for NO in more proximal metabolic reprogramming than the TCA break we carried out detailed $^{13}C$ tracing experiments. $^{1}H$-{$^{13}C$} heteronuclear single-quantum coherence (HSQC) nuclear magnetic resonance (NMR) spectra confirmed glycolytic shifts (Fig. 3a and Supplementary Fig. 3A) and showed that labeled TCA cycle-derived metabolites (e.g., Glu and Asp) were higher in $Nos2^{-/-}$ compared to WT macrophages, suggesting NOS2-dependent suppression of $^{13}C$-glucose entry into the TCA during polarization. Furthermore, NMR revealed amounts of $^{13}C$ labeled itaconate in $Nos2^{-/-}$, consistent with the notion that cells lacking NOS2 have both intact TCA and active cis-aconitate decarboxylase.

Deeper analysis of $^{13}C$ by MS revealed more profound effects of NO on carbon flow. We observed reduced $m + 2$ citrate enrichment in stimulated WT compared to $Nos2^{-/-}$ macrophages (Fig. 3b). Mass isotopologue distribution (MID) of citrate refers to $m + 2$ and $m + 3$ isotopologues as results of the initial entry of glucose into the TCA via pyruvate dehydrogenase (PDH) or pyruvate anaplerosis, respectively. $m + 4$, $m + 5$ and $m + 6$ isotopologues indicate $^{13}C$-glucose-derived carbons that have cycled two or more times through the TCA (Supplementary Fig. 3B). In fact, fractional enrichment of $^{13}C$ into $m + 2$ citrate in $Nos2^{-/-}$ was largely unaffected by stimulation whereas in WT ~90% of citrate remained unlabeled ($m + 0$), indicating that it was not derived from glucose. This differential labeling in WT vs. $Nos2^{-/-}$ was reflected in other TCA metabolites (Fig. 3c–i and Supplementary Fig. 3A). In addition, $Nos2^{-/-}$ macrophages had higher levels of $m + 4$ isotopologues demonstrating higher glucose oxidation through multiple rounds of an intact TCA (Fig. 3c–i).

Interestingly, $Nos2^{-/-}$ showed comparable enrichment in $m + 3$ citrate, cis-aconitate and malate to that seen in WT cells (Fig. 3b, c, i). $m + 3$ citrate and $m + 3$ malate derive from $m + 3$ oxaloacetate (OAA) produced by pyruvate carboxylation followed by condensation with unlabeled acetyl-CoA (citrate) or free exchange via MDH (malate). Surprisingly WT macrophages had more $m + 3$ aspartate (Supplementary Fig. 3C), produced by transamination of OAA (Fig. 1d). Of note, we found higher induction of pyruvate carboxylase (PC, *Pcx*) during stimulation of WT compared to $Nos2^{-/-}$ (Supplementary Fig. 3D, E). This suggest that alternative pathway promoting anaplerosis through PC may be upregulated. Consistent with functional aconitase, $m + 3$ isotopologues downstream of cis-aconitate were increased in $Nos2^{-/-}$ (Fig. 3e–h).

These data suggest NO-dependent blockage of glucose oxidation via PDH- but not PC-initiated TCA. Thus, we calculated the ratio of $m + 2$ citrate/$m + 3$ pyruvate to normalize for upstream

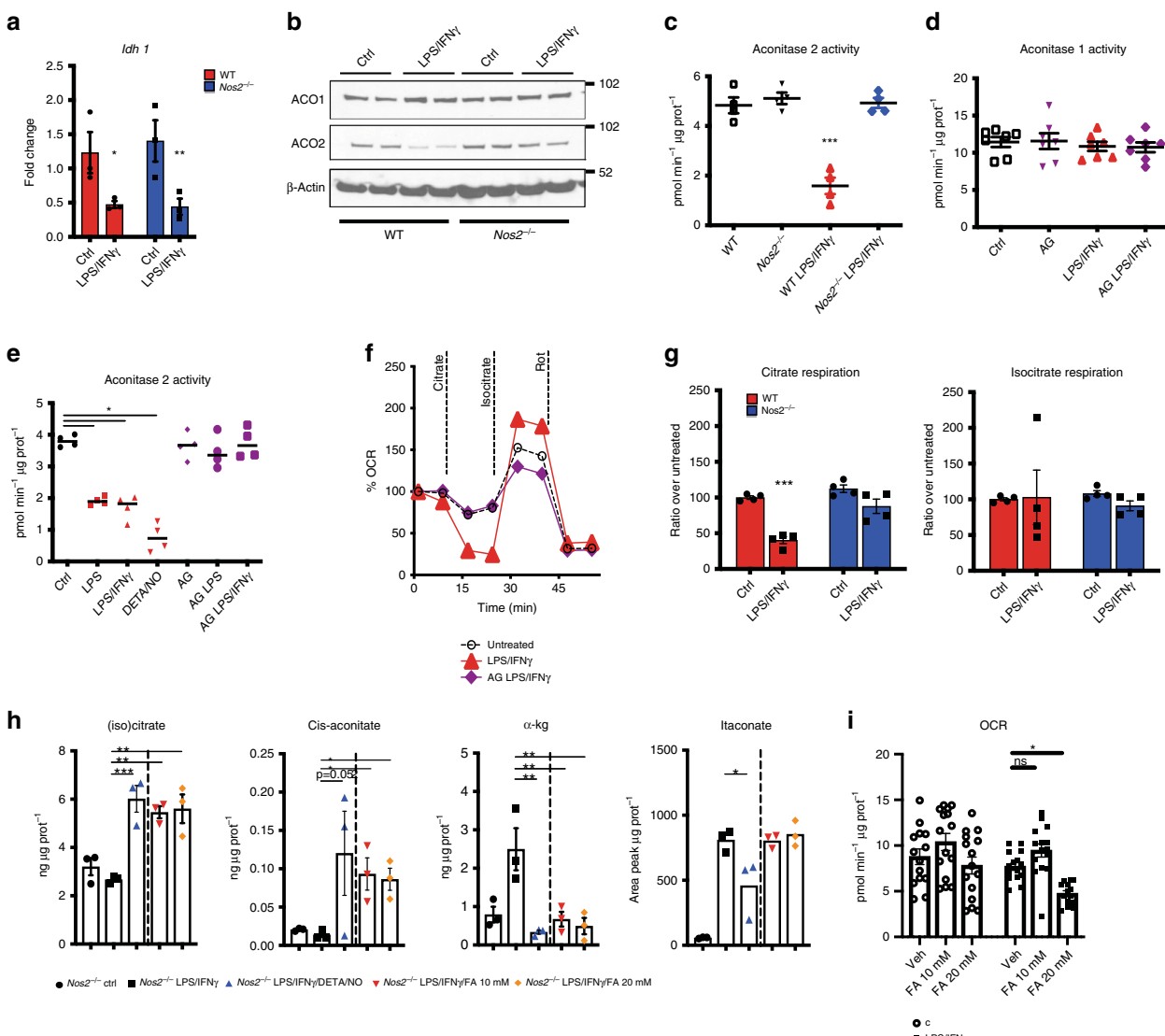

**Fig. 2 The TCA Break is due to NO targeting of mitochondrial aconitase. a** WT and *Nos2*−/− BMDMs were stimulated for 8 h and mRNA was extracted from total cell lysates and analyzed by qPCR for *Idh1* (*n* = 6) (two-way ANOVA, Sidak's post-tests; *p*-values = 0.033, 0.0093). **b** Whole-cell lysates of macrophages derived from two independent WT and *Nos2*−/− mice were analyzed by western blot for cytosolic and mitochondrial aconitase (ACO 1-2). β-actin was used as loading control. **c, d** Aconitase 2 and total aconitase (aconitase 1) enzymatic activities in WT vs. *Nos2*−/− BMDMs stimulated over night (O.N.). Where indicated cells were treated with NOS2 inhibitor Aminoguanidine (AG) 1 h prior to stimulation (*n* = 6). **e** Damaging effect of endogenously produced or exogenously provided NO (DETA/NO) on activity of aconitase 2 (*n* = 3). **f** Representative Seahorse analysis of oxygen consumption rates (OCR) in permeabilized WT and AG-pretreated BMDMs stimulated O.N. Citrate, tartronate, ADP and PMP were co-injected (first event marker). Isocitrate was injected at the second event marker and rotenone (rot) was injected lastly. **g** Quantifications (percentage relative to ctrl cells) of exogenous citrate and isocitrate-dependent state 3 OCR in WT vs. *Nos2*−/− (*n* = 6). Data were analyzed by one-way (**c–e**) or two-way ANOVA (**f, g**), (*p* < 0.0001) with Tukey's post-tests. **h** *Nos2*−/− BMDMs were stimulated with LPS + IFNγ in the presence of either DETA/NO (500 μM) or FA. Metabolites were quantified by ESI-LC/MS-MS and are reported as normalized total ng or peak area. Data (*n* = 6) were analyzed by one-way ANOVA with Dunnett's post-tests. **i** Bar graphs showing quantified, protein normalized, basal OCR from stress tests of *Nos2*−/− macrophages treated with vehicle or FA 1 h prior to O.N. stimulation. Data (*n* = 3) were analyzed by two-way ANOVA with Sidak's post-tests. All error bars display mean ± SEM. Source data are provided as a Source Data file.

changes in glycolysis[13] and found low ratios only in stimulated WT (Fig. 3j). This finding, together with decreased acetyl-CoA availability (Supplementary Fig. 3F), confirmed NOS2-dependent decreased pyruvate flux through PDH. Moreover, we found that LPS/IFNγ-induced reductions in pyruvate-elicited respiration were dependent on NOS2 and NO (Fig. 3k and Supplementary Fig. 3G).

**Decreased carbon flux through PDH is Hif1α independent.** Decreases in pyruvate oxidation have been suggested to result

from HIF1α-mediated pyruvate dehydrogenase kinase 1 (Pdk1) induction, with subsequent phosphorylation and inactivation of PDH by PDK1[32–34]. Since we did not observe decreased *m* + 2 citrate in macrophages lacking NOS2, we investigated possible NO effects on HIF1α. We found that stimulation resulted in increased *Hif1α* and *Pdk1* levels in both WT and *Nos2*−/− macrophages (Fig. 4a and Supplementary Fig. 4A, B). Furthermore, we found considerable levels of PDK1 and phospho-PDK1 in resting cells, consistent with high-glycolytic status of murine BMDMs[35], and these levels remained unchanged after

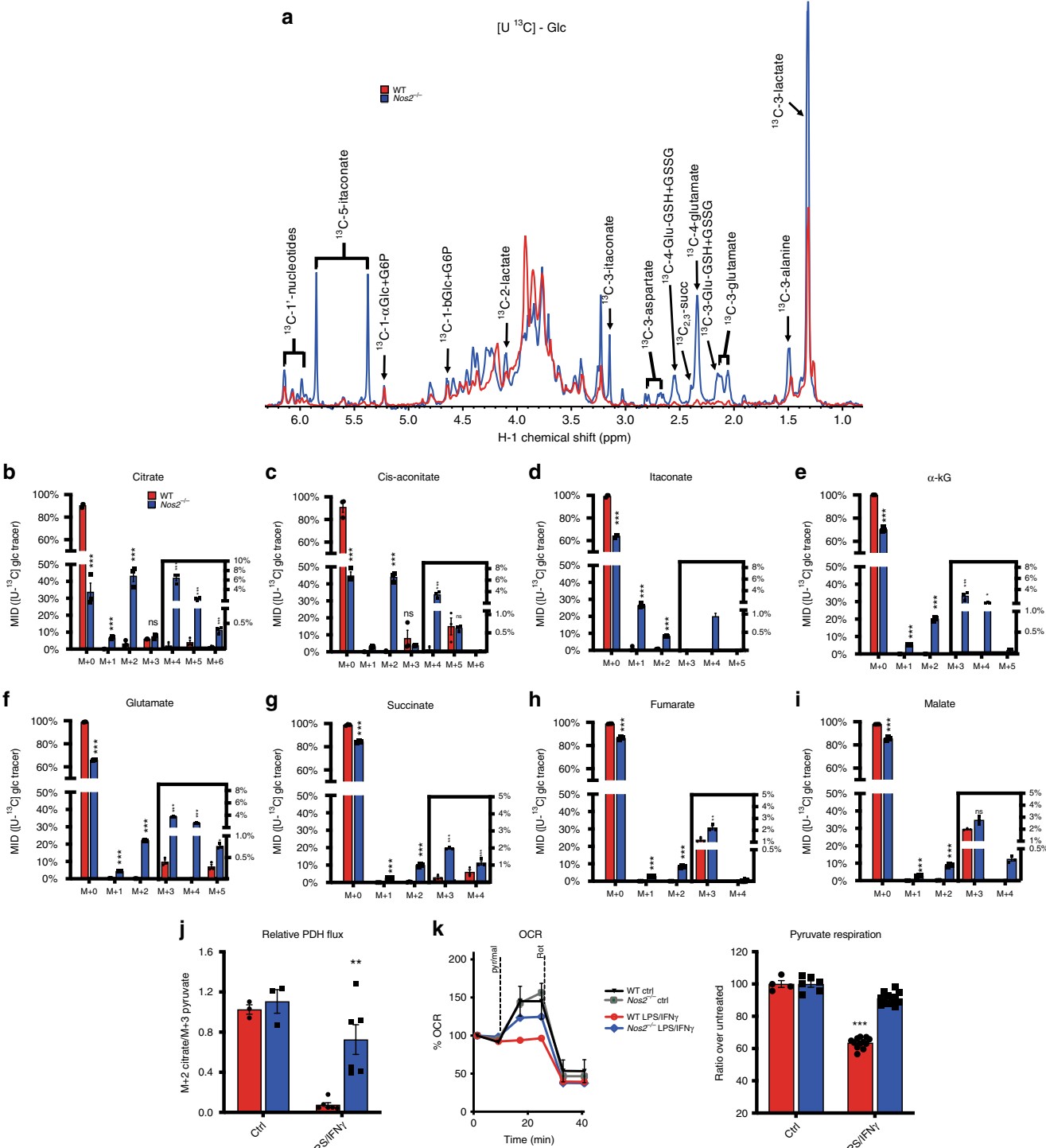

**Fig. 3 Absence of NO promotes pyruvate oxidation via PDH. a** Polar extracts from WT and $Nos2^{-/-}$ BMDMs cultured with [U-$^{13}$C] glucose were analyzed by 1D $^1$H{$^{13}$C}_HSQC NMR. Spectral comparisons of stimulated cells are superimposed and show labeled TCA-derived metabolites in WT (red) vs. $Nos2^{-/-}$ (blue). Spectra are representative of two experiments ($n = 6$). **b–i** MID from GC-MS analysis of citrate, cis-aconitate, itaconate, α-kG, glutamate, succinate, fumarate, and malate. $m + 0$ to $m + 6$ reveal the contribution of glucose-labeled metabolites when TCA cycle can run multiple turns. Data ($n = 6$) were analyzed by two-way ANOVA (interaction < 0.0001) (Sidak's post-tests). **j, k** show flux through PDH. **J** Ratio of $m + 2$ citrate/$m + 3$ pyruvate indicating relative oxidation through PDH. Bar graphs represent pooled data ($n = 6$) and were analyzed by two-way ANOVA (interaction $p = 0.0075$) with Tukey's post-tests. **k** Representative Seahorse analysis of permeabilized BMDMs where state 3-OCR was elicited by a mixture of pyruvate and malate to measure PDH flux. Bar graphs show quantified respiration in O.N. activated WT and $Nos2^{-/-}$ BMDM ($n = 9$) as percentage of OCR relative to untreated. Data ($n = 6$) were analyzed by two-way ANOVA (interaction < 0.0001) (Sidak's post-tests). All error bars display mean ± SEM. Source data are provided as a Source Data file.

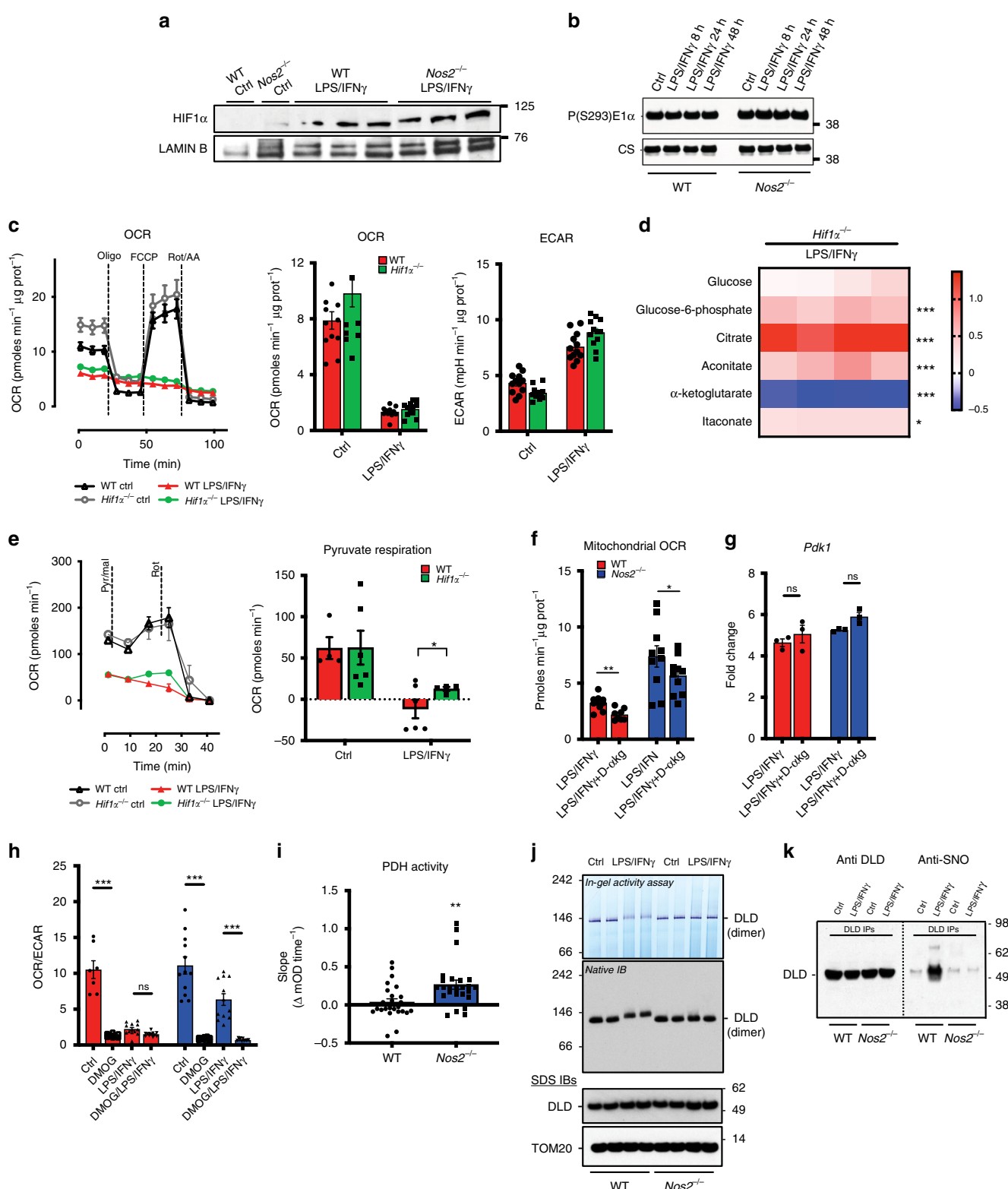

stimulation (Supplementary Fig. 4C). Accordingly, activation did not change phosphorylation status of PDH (Fig. 4b) and treatment with PDK inhibitor dichloroacetate (DCA) had no effect on OCR (Supplementary Fig. 4D). In addition, OCR decrease in M1-polarized $Hif\alpha^{-/-}$ macrophages was comparable to that of WT, and we detected equivalent ECAR levels in WT and $Hif1\alpha^{-/-}$ macrophages (Fig. 4c), despite HIF1α-mediated regulation of glycolytic genes (Supplementary Fig. 4E, F). Metabolomic

profiling of $Hif\alpha^{-/-}$ BMDMs showed levels of LPS/IFNγ-induced glycolytic intermediates, citrate, itaconate and α-KG comparable to WT, indicating that $Hif1\alpha^{-/-}$ BMDMs still have a block in the TCA at ACO2 (Fig. 4d). In further support of this, we found citrate-fueled respiration equally impaired in $Hif1\alpha^{-/-}$ and WT macrophages (Supplementary Fig. 4G). Pyruvate-dependent respiration was still compromised in $Hif1\alpha^{-/-}$ cells (Fig. 4e), and PDK1 protein was unaffected (Supplementary Fig. 4H). Lastly, we

**Fig. 4 Decreased carbon flux through PDH is Hif1α independent. a** Immunoblot (IB) for HIF1α protein of three independent nuclear extracts from WT and $Nos2^{-/-}$ BMDMs stimulated for 24 h. Lamin B was used as loading control ($n = 6$). **b** IB for PDH-Phospho-Ser[293] in mitochondrial extracts; citrate synthase (CS) was used as loading control ($n = 6$). **c** O.N.-treated WT and $Hif1α^{-/-}$ BMDMs were seeded in Seahorse XF96 cell culture plates and sequential treated with oligomycin (Oligo), FCCP, and rotenone plus antimycin A (Rot/AA). Basal OCR and extracellular acidification rate (ECAR) are quantified ($n = 3$). Data were not significant by two-way ANOVA with Sidak's post-tests ($p$-values = 0.59, 0.1675). **d** Heat-maps from GC-MS analysis of metabolites of glycolysis and TCA cycle from $Hif1α^{-/-}$ BMDMs. Data show log10 ratio from the average peak areas of stimulated cells compared to unstimulated. Data ($n = 4$) were analyzed by two-way ANOVA (interaction < 0.0001) (Sidak's post-tests). **e** Representative Seahorse analysis of pyruvate/malate respiration in WT and $Hif1α^{-/-}$ BMDMs. Bar graphs show quantified OCR ($n = 3$) (two-way ANOVA, interaction $p = 0.036$, Sidak's post-tests). **f** Protein normalized basal OCR from stress tests of WT and $Nos2^{-/-}$ macrophages treated with Dimethyl α-KG (D-αKG) 1 h prior to O.N. stimulation. Data were analyzed by Student's $t$-test ($n = 3$). **g** BMDMs as in **f** and stimulated for 8 h. mRNA from total cell lysates was analyzed by qPCR for *Pdk1*. Data ($n = 6$) were not significant by two-way ANOVA (interaction $p = 0.74$) (Sidak's post-tests). **h** Basal OCR/ECAR in WT and $Nos2^{-/-}$ BMDMs stimulated O.N. and pretreated with DMOG. Data ($n = 3$) were analyzed by two-way ANOVA (interaction < 0.0001) (Sidak's post-tests). **i** Enzymatic activity of PDH in total cell lysates from WT and $Nos2^{-/-}$ BMDMs after O.N. activation. Data ($n = 6$) were analyzed by unpaired $t$-test with Welch's correction ($p = 0.0038$). **j** PDH-E3 (DLD) in-gel activity assay and IB on native gels of mitochondrial fractions from macrophages from two independent WT and $Nos2^{-/-}$ mice. **k** Mitochondrial fractions from control and LPS + IFNγ stimulated WT and $Nos2^{-/-}$ BMDMs were used as inputs to immunoprecipitate DLD. Anti DLD and Anti-CysSNO IBs were performed. All error bars display mean ± SEM. Source data are provided as a Source Data file.

---

pharmacologically manipulated HIF1α in WT and $Nos2^{-/-}$ macrophages with dimethyl α-KG (Dα-KG) or dimethyloxalylglycine (DMOG). Treatment with Dα-KG, a prolyl hydroxylase (PHD) agonist, did not improve respiration (Fig. 4f) and had no effect on glycolysis (Supplementary Fig. 4I) or expression of Pdk1 or Pcx (Fig. 4g and Supplementary Fig. 4J). Interestingly, direct targeting of PHDs via the inhibitor DMOG, promoting HIF1α stabilization, strongly skewed OCR/ECAR towards glycolysis in resting WT and $Nos2^{-/-}$ (Fig. 4h), suggesting that boosting the amount of HIF1α increases glycolytic activity by affecting Pdk1 expression regardless of activation (Supplementary Fig. 4K). Finally, when fueling with pyruvate, OCR after DMOG was similar to vehicle, confirming that inhibition of pyruvate respiration in activated macrophages is solely dependent on NO production (Supplementary Fig. 4L).

Having excluded the Hif1α-mediated mechanism of PDH inhibition, we tested whether this was the result of product inhibition due to blockade of ACO2. We found that this was not the case as fluoroacetate (FA)[31] did not affect pyruvate respiration (Supplementary Fig. 4M). We then hypothesized that PDH might be directly inhibited in the presence of NO. Therefore, we assessed PDH enzymatic activity and found nearly undetectable activity in stimulated WT cells (Fig. 4i), so we used native in-gel activity assays to further investigate the mechanism. These data revealed decreased activity of the PDH-E3 subunit (dihydrolipoyl dehydrogenase, DLD) accompanied by shifts in in-gel migration in activated WT compared to $Nos2^{-/-}$ macrophages (Fig. 4j), suggesting the presence of post-translational modification. We then immunoprecipitated DLD and detected strong reactivity in immunoblots (IBs) for anti Cysteine-SNO only in activated WT cells (Fig. 4k).

Taken together, these data demonstrate that during polarization Hif1α does not control macrophage mitochondrial metabolic reprogramming via PDK-mediated modulation of PDH activity, rather NO-dependent suppression of PDH occurs, likely via S-nitrosation of DLD.

**LPS/IFNγ increase glutamine utilization in NO-dependent manner.** Having demonstrated NO-dependency of decreased pyruvate oxidation through PDH and disruption of TCA cycle (Fig. 3, 4), we next asked whether the accumulation of citrate and other TCA metabolites (Fig. 1c) might reflect utilization of carbon sources other than glucose. Since glutaminolysis is a major anaplerotic pathway in macrophages[36], we tested if this might be activated to compensate for ACO2 and PDH malfunction in stimulated cells. Measurement of extracellular glutamine levels

revealed NOS2-dependent uptake of glutamine (Fig. 5a and Supplementary Fig. 5A, B). [U-13C]-glutamine tracing showed that, despite comparable upregulation of glutaminase (Fig. 5b and Supplementary Fig. 5C), 13C fractions of glutamate, succinate and citrate were smaller in macrophages lacking NOS2 (Fig. 5c), which displayed higher fraction of unutilized [U-13C] glutamine (Supplementary Fig. 5D). These data indicate a major contribution of glucose to metabolite pools in macrophages lacking NOS2. Moreover, in WT cells 51% of the succinate was labeled but only ~10% of fumarate contained 13C, suggesting that only small amounts of 13C succinate transit via SDH and that alternative routes yield fumarate and malate in stimulated WT[26,37]. In contrast, in $Nos2^{-/-}$ cells a higher percentage of carbon transited through SDH (Fig. 5c). Blocking ACO2 with FA, in absence of NO, promoted increased glutamine utilization and elicited a greater enhancement of OCR in the presence of glutamine (Fig. 5d and Supplementary Fig. 5E), confirming switch toward glutaminolysis to sustain mitochondrial metabolism when glucose-derived substrates are lacking.

Taken together, these data demonstrate that during LPS/IFNγ treatment the presence of NO induces uptake of glutamine and its utilization due to the cessation of glucose flux via PDH and ACO2. Concomitantly, SDH activity falls in an NO-dependent manner and carbon is re-routed. These data emphasize the ability of macrophages to adapt to NO production through exploitation of alternative pathways to satisfy the need for bioenergetics substrates.

**NO inhibits mitochondrial ETC complexes and promotes their loss.** Older studies had suggested that NO might directly target and inactivate mitochondrial respiratory chain[21,24,38]. Therefore, having ruled out NO effects on mitochondrial number per se (Supplementary Fig. 6A), we performed native IBs to assay the integrity of ETC (Fig. 6a). Our analysis revealed profound stimulation-dependent reduction of supercomplexes containing CI plus dimerized CIII, levels of CIII, and free CI, CII, and CIV in WT but not in $Nos2^{-/-}$ macrophages (Fig. 6a). Next, we performed sodium dodecyl sulfate (SDS) IBs and detected NOS2-dependent reductions in CI (NDUFS1), CII (SDHB), CIII (UQCRC2), and CIV (MTCO1) subunits (Fig. 6b), which suggest that changes in abundance of supercomplexes are due to changes in protein expression and not simply alterations in supercomplex formation.

In-gel NADH-dehydrogenase assays confirmed Nos2-dependent decline in CI activity in BMDMs after stimulation (Fig. 6c). Similarly, CII and CIV activities were progressively

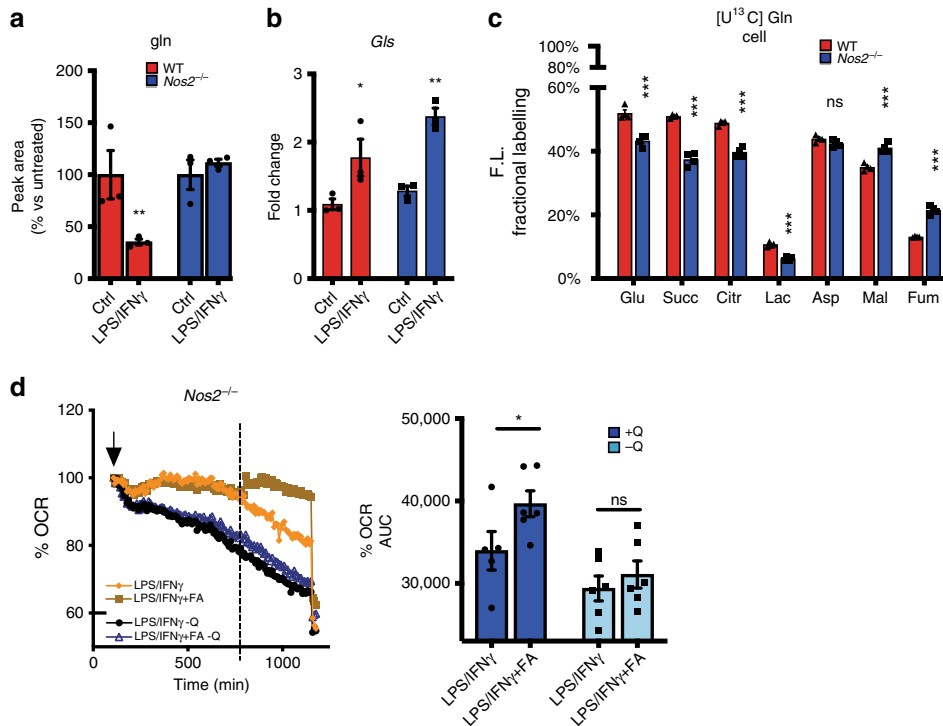

**Fig. 5 LPS/IFNγ increase glutamine utilization in NO-dependent manner. a** Extracellular glutamine levels were quantified by GC-MS in O.N. stimulated WT vs. *Nos2*⁻/⁻. Data (n = 9) were normalized to the absolute peak area of the metabolite in WT control and were analyzed by two-way ANOVA (interaction p = 0.0037) (Sidak's post-tests). **b** RNA as in **2a** were analyzed for *Gls* expression (two-way ANOVA with Sidak's post-tests, p-values = 0.0114, 0.0024). **c** Fractional contribution of ¹³C-labeling from [U-¹³C] Glutamine in metabolites of the TCA in activated WT vs. *Nos2*⁻/⁻ BMDMs Data (n = 4) were analyzed by two-way ANOVA (interaction < 0.0001) (Sidak's post-tests). **d** OCR in *Nos2*⁻/⁻ BMDMs after addition of LPS + IFNγ (at the arrow indicated) in the presence or absence of glutamine (Q). Bar graphs showing quantification of OCR after addition of FA compared to vehicle (dotted line). Data (n = 3) were analyzed by two-way ANOVA (interaction p = 0.0192) (Sidak's post-tests). All error bars display mean ± SEM. Source data are provided as a Source Data file.

decreased, whereas they remained unchanged in *Nos2*⁻/⁻ (Fig. 6d, e). Furthermore, stimulated *Nos2*⁻/⁻ cells treated with the NO donor had dysfunctional CI and CII (Fig. 6f, g). Moreover, protein levels and activity of CI and CII, as well as CI to CIV of WT macrophages is rescued completely by the presence of AG before or halfway through stimulation (Supplementary Fig. 6B). Altogether these experiments proved concretely the link between NO and macrophage ETC impairment.

The concomitant decrease in the activity of CI, CII and CIV together with the intact OCR of stimulated WT macrophages upon direct substrate feeding (Fig. 2f, g and Supplementary Fig. 6C, D), suggest that if were electron donors available, remaining ETC complexes are sufficient to compensate for NO-dependent loss.

**NO directs metabolic reprogramming in vivo**. To address whether, similar to that observed in activated BMDMs in vitro, nitric oxide was involved in macrophage metabolic reprogramming in inflammatory conditions in vivo we assessed metabolism during a modified Schwartzman reaction[39].

Mice were given thioglycolate intraperitoneally (i.p.) 3 days before priming with IFNγ, then challenged with LPS (Fig. 7a). This reaction yielded increases in nitrite and citrulline in the peritoneal lavage (Fig. 7b and Supplementary Fig. 7A) indicating substantial iNOS activity. Moreover, we confirmed that like BMDMs in vitro, during the Schwartzman reaction, peritoneal CD11b⁺ Ly6G⁻macrophages were prominent producers of citrulline (Fig. 7c), accumulated itaconate (Fig. 7d) and had

decreased ornithine (Fig. 7e). Furthermore, consistent with in vitro findings, CD11b⁺ Ly6G⁻macrophages from the peritoneum showed increased citrate to isocitrate ratios (Fig. 7f and Supplementary Fig. 7B) and reduced α-KG (Fig. 7g and Supplementary Fig. 7C) indicative of ACO2 blockade. Accordingly, we observed decreased aconitase 2 activity in total CD11b + cells isolated during peritoneal inflammation confirming this relationship (Fig. 7h).

Although these parameters recapitulated many in vitro findings, in vivo macrophages did not show increases in absolute intracellular quantities of citrate during the Shwartzman reaction (Supplementary Fig. 7D). However, citrate increased in the lavage fluid (Fig. 7i) as did citrate/α- KG ratios and itaconate; ornithine levels dropped. In vivo macrophages also showed indications of anaplerosis, with glutamine influx and glutamate usage, despite apparent highly controlled absolute levels of glutamine and glutamate in the lavage (Fig. 7j and Supplementary Fig. 7E). Remarkably, we found that all these characteristics were dependent on Nos2, as lavage from *Nos2*⁻/⁻ mice under these same conditions not only displayed increased arginine metabolites apart from Nos2 activity (Fig. 7i and Supplementary Fig. 7E), but also excessive itaconate and no increase in citrate (Fig. 7i). Intracellularly, peritoneal *Nos2*⁻/⁻ macrophages showed absence of break at ACO2, restored succinate and malate levels, and weaker glutamine-derived carbon usage (Fig. 7j).

Finally, we analyzed the function of CI and CII components of mitochondrial ETC of CD11b⁺ cells isolated from mice undergoing Shwartzman reaction and found that these were

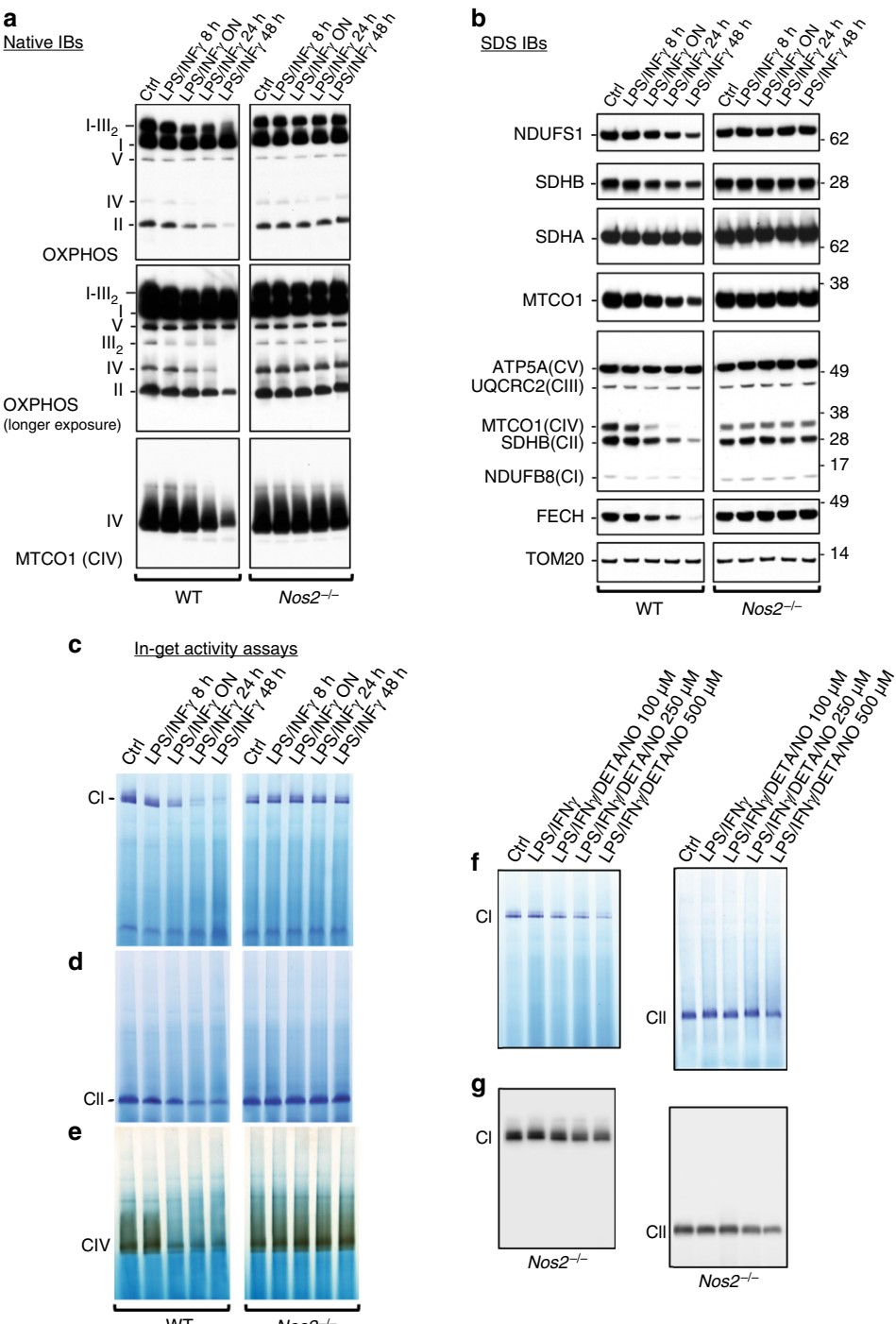

**Fig. 6 NO inhibits mitochondrial ETC complexes and promotes their loss. a** Native IBs were performed to assess assembly of complexes I–V (CI–V) in BMDMs after activation with LPS + IFNγ for indicated times. Total OXPHOS IB was performed to detect the CI subunit NDUFB8, the CII subunit SDHB, the CIII-Core protein 2 (UQCRC2), the CIV subunit I (MTCO1, also detected by IB with specific antibody) and the CV alpha subunit (ATP5A). **b** SDS IBs including the CI subunit NDUFS1, the CII subunit SDHA, and the Fe-S protein ferrochelatase (FECH). Levels of TOM20 were used as mitochondrial matrix marker for loading controls. IBs were performed on samples described in **a**. (**a–d**, $n = 4$ biological replicates). **c** CI, **d** CII, and **e** CIV in-gel activity assays were performed in activated WT and $Nos2^{-/-}$ cells at reported time points. **f** CI and CII in-gel activities in $Nos2^{-/-}$ BMDMs stimulated with LPS + IFNγ in the presence of increasing concentration of DETA/NO (100, 250 and 500 μM) (**g**) SDS IBs for CI (NDUFS1) and CII (SDHA) performed on samples described in **f**.

considerably compromised in NOS2-dependent manner, similar to BMDMs (Fig. 7k).

Taken together, these data show that not only do many characteristics of macrophage metabolic rewiring recapitulate in vivo, but these are almost universally dependent on NO.

## Discussion

Glycolytic commitment, characterized by downregulation of OXPHOS and ATP production by mitochondria is a key feature in the response of macrophages and DCs to proinflammatory stimuli[40]. Current models suggest that rewiring of carbon utilization

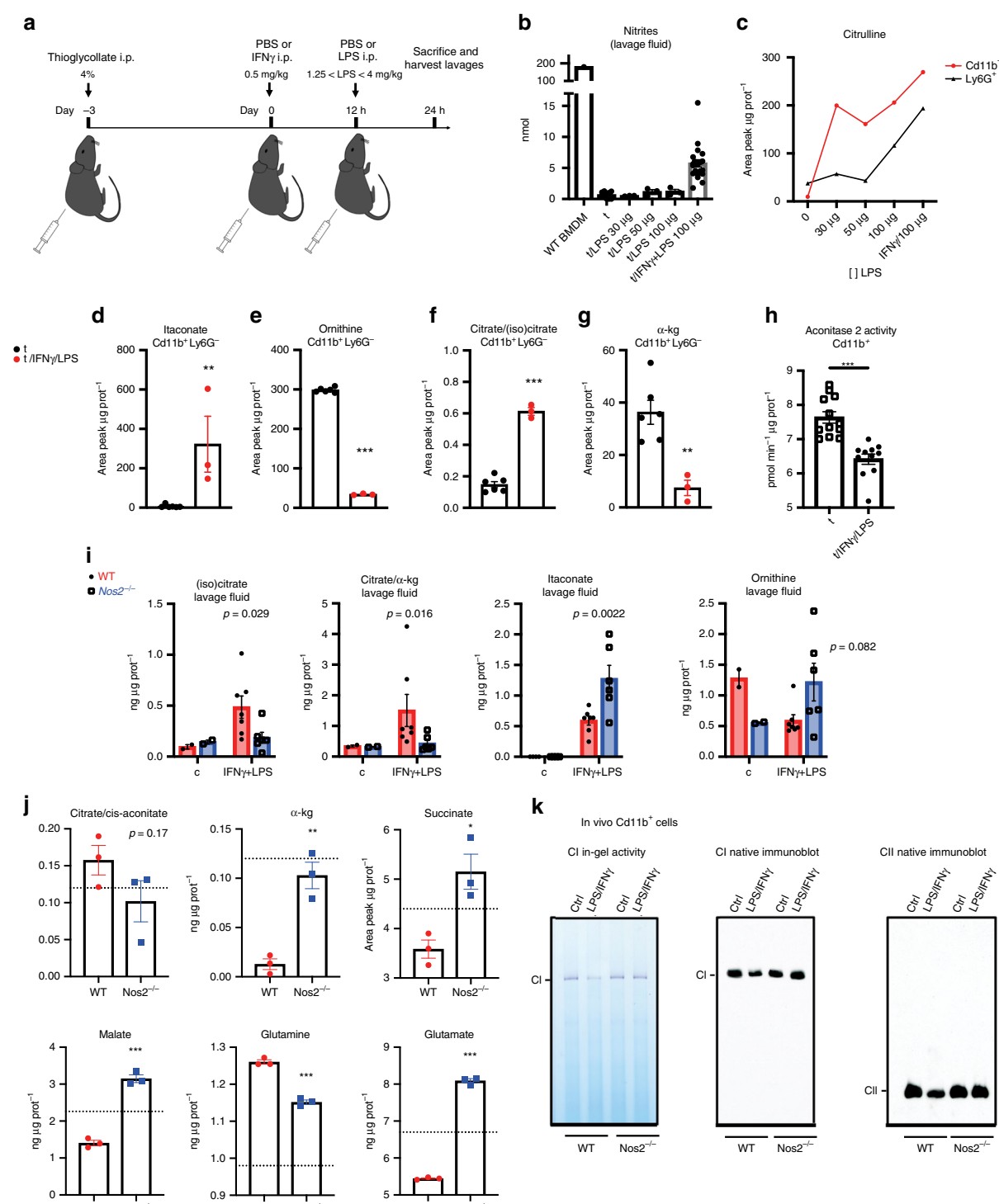

takes place through the coordinated suppression of IDH1, SDH, and the PDH complex[4,10,11,26,32]. This model predicts that macrophages primarily convert glucose-derived pyruvate to lactate, while anaplerotic glutaminolysis facilitates succinate accumulation, which yields Hif1α stabilization, leading to IL1β production and suppression of PDH via Pdk1[2,4,10,11,26,32]. Our study calls for reconsideration of this proposed model. We provide evidence for NO being the main orchestrator of changes in macrophage mitochondrial metabolism during exposure to inflammatory stimuli (Fig. 8).

One of the most profound macrophage alterations is reduction in carbon flow from citrate to α-KG and concomitant decline in

α-KG levels. Jha et al. suggested that this break in the TCA was due to transcriptional downregulation of *Idh1*[26]. However. we demonstrate that lack of carbon continuity between citrate and α-KG and declines in α-KG are results of NO production. We find that although NOS2 is required for alterations in α-KG homeostasis, it is dispensable for reductions in Idh1 mRNA. IDH1 is a cytoplasmic enzyme and less likely to direct affect TCA function. In contrast, multiple lines of evidence suggest that ACO2 is the source of TCA break. Unlike IDH1, ACO2 is a known target of NO attack[22,24,30], and we document reductions in ACO2 protein and activity during polarization. Most importantly, our studies show that citrate-mediated respiration is blunted in M1

**Fig. 7 NO directs metabolic reprogramming in vivo. a** Mice were injected i.p. with thioglycolate and challenged after 3 days with injections of IFNγ or PBS i.p. 12 h later, mice were administered LPS i.p. and euthanized in the subsequent 12 h. Peritoneal lavage fluid was harvested and cells isolated: both processed for metabolic studies. **b** Nitrite levels were quantified by Griess reaction performed on concentrated lavage fluid in comparison with $2 \times 10^6$ LPS/IFNγ stimulated BMDM-derived culture media. Results are shown as total nmol. ($n > 3$ for each group). **c** Mice were challenged as in **a** and either Ly6G + or Cd11B + cells were isolated from peritoneal lavage and subjected to metabolite extraction and subsequent ESI-LC/MS-MS analysis. Normalized peak area of citrulline is shown. **d–g** Quantified metabolites in Cd11b + Ly6G- isolated from WT mice challenged for Shwartzman reaction ($n = 6$) ("t" represents mice injected with thioglycolate alone). **h** Aconitase 2 activity in total CD11b$^+$ cells from mice undergoing Shwartzman reaction ($n = 3$). Data were analyzed by Student's $t$-test. **i** WT and $Nos2^{-/-}$ mice were challenged as in **a** and peritoneal lavage was concentrated, extracted and analyzed by ESI-LC/MS-MS analysis. Absolute amounts as ng/total μg protein in lavage fluid are shown. Data were analyzed by two-way ANOVA with Sidak's post-tests ($n = 8$). Shown $p$-values indicate WT vs. $Nos2^{-/-}$ comparisons in IFNγ + LPS condition. **j** Quantified metabolites in Cd11b$^+$Ly6G$^-$ isolated from challenged WT and $Nos2^{-/-}$ mice. The dotted line represents the average level of metabolite in thioglycolate alone-group. Data were analyzed by Student's $t$-test ($n = 3$). **k** CI in gel activity and SDS IBs for CI (NDUFS1) and CII (SDHA) in total CD11b$^+$ cells from challenged WT vs. $Nos2^{-/-}$ mice ($n = 2$). Source data are provided as a Source Data file.

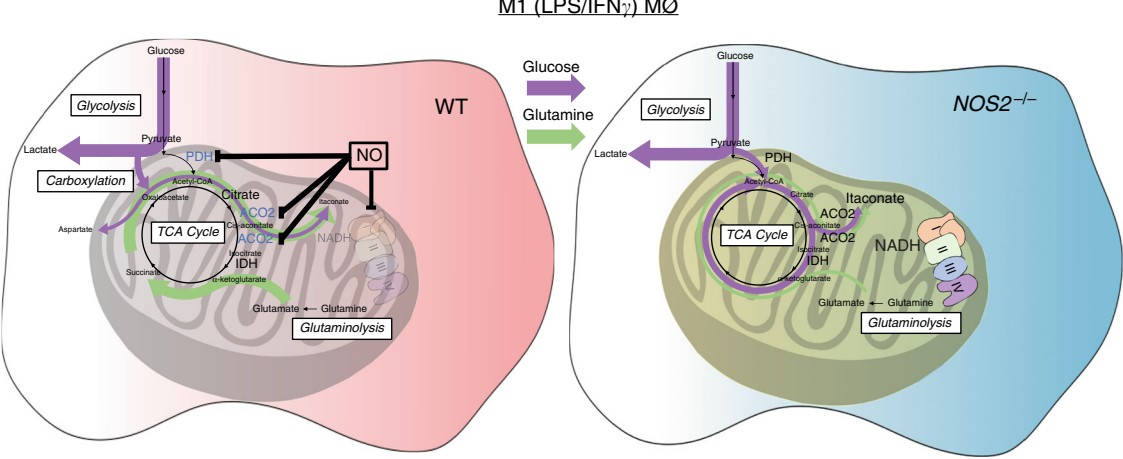

**Fig. 8 NO orchestrates metabolic reprogramming in M1 macrophages.** Macrophage activation in the presence of Nitric Oxide (NO) results in multiple metabolic rewirings. Aconitase 2 is inhibited and entrance of carbon into TCA cycle via pyruvate dehydrogenase is halted. In turn, compensatory carboxylation and glutaminolysis are enhanced, leading to citrate accumulation and limited itaconate. Because of the breaks, the lack of NADH and reduced substrates leads to inactive mitochondrial complexes. In $Nos2^{-/-}$ BMDMs glycolytic flux is maintained and carbon passage through PDH and ACO2 is intact. In this scenario, production of itaconate is increased and mitochondrial complexes display full functional activity. The arrows represent the general direction of the metabolic flow in the system with the specific contribution of glucose (purple) and glutamine (green). PDH pyruvate dehydrogenase, ACO2 mitochondrial aconitase, IDH isocitrate dehydrogenase, LPS lipopolysaccharide, IFNγ interferon γ.

macrophages whereas isocitrate respiration is intact, definitively placing the break at ACO2 not IDH1. This is the first demonstration that ACO2 is the metabolic target in BMDMs. In addition, specific inhibition of ACO2 is in accordance with known lipophilic nature of NO chemistry[22].

Although alterations in TCA are considered to be vital for appropriate immune response[2,9] it is interesting to note that, as reported also by others[41–43], $Nos2^{-/-}$ macrophages maintained full inflammatory phenotype, as indicated by higher inflammatory gene expression and cytokine secretion than WT. Our findings suggest that in $Nos2^{-/-}$ cells, a functional TCA and better stabilization of Keap1 (Kelch-like ECH-associated protein 1), regulator of Nrf2, could prevent Nrf2-mediated inhibition of transcription. Nrf2 is known to attenuate inflammation in myeloid cells[44], and our data suggest that the presence of NO reprograms macrophage metabolism while promoting protective processes to mitigate NO-induced damage, at the expense of full inflammatory responses. This is in line with the impact of NO in DCs in limiting adaptive immune cell activation[42].

We demonstrate that carbon flux through the PDH complex is severely compromised in M1 macrophages. Reports have suggested that in polarized macrophages, pseudo-hypoxia, caused by reduction in α-KG /succinate ratios, results in Hif1α stabilization and expression of Pdk1, that phosphorylates PDH, reducing pyruvate oxidation and promoting Warburg effect[2,4,32,34,45]. In

contrast, we support a different mechanism for this phenomenon. Although $^{13}$C tracing studies do show cessation of glucose carbon flux through PDH, we find this effect to be dependent on NO production, while Hif1α stabilization, PDK accumulation, and PDH phosphorylation are independent of NOS2. Moreover, PDK1 expression, the TCA break and compromised PDH activity were all evident in macrophages lacking Hif1α, consistent with the data from Prados et al.[46]. Our data suggest that NO directly targets PDH as we found the DLD subunit to be inactive and positive for cysteine-nitrosation in NOS2-dependent manner. This hypothesis is supported by reports of nitrosation of the PDHE3 subunit by nitroxyl or peroxynitrite[47,48]. Therefore, instead of Hif1α/PDK-mediated effect, we propose that $N_2O_3$ or nitrogen dioxide $NO_2$, from reaction of NO with oxygen, directly inactivates PDH. The fact that DLD is targeted by NO has implications due to its association with multiple metabolic enzymes, including oxoglutarate dehydrogenase (OGDH) and branched chain alpha keto acid dehydrogenase (BCKDH). Thus, we predict that OGDH and BCKDH, like PDH, are stalled due to NO-mediated injury. This requires further biochemical dissection but suggests that additional NO-mediated rewiring events are yet to be discovered.

NO affects the accumulation of metabolites hallmarks of M1 polarization: citrate, succinate and itaconate. Our data support a model in which, during stimulation, glucose uptake increases,

Irg1 is induced and citrate and itaconate are synthesized incorporating glucose-derived carbon. When NO levels rise, PDH suppression limits further utilization of glucose carbon resulting in increased compensatory glutamine influx that routes carbon through TCA to yield citrate and itaconate. As NO levels suppress ACO2 activity, citrate accumulates and is exported, whereas itaconate production is limited due to the lack of cis-aconitate. This is supported by the very rapid detection of itaconate[14], the higher levels of itaconate in $Nos2^{-/-}$, the sensitivity of PDH and ACO2 to direct NO inhibition, and the substantial incorporation of glucose carbon into itaconate in $Nos2^{-/-}$, whereas in WT it is largely unlabeled. Importantly, while availability of cis-aconitate supports production of itaconate, Irg1 only negligibly affects cis-aconitate compared to the effect elicited by NO on ACO2. Moreover, GABA shunt itself may represent an effort to bypass suppression of OGDH.

Our data demonstrate that NO leads to the decline in the levels of mitochondrial ETC complexes, a phenomenon that has not been fully described and often overlooked in M1 macrophages. We confirmed that NO is sufficient and iNOS is required.

Our experiments with permeabilized cells suggest that when directly fueled in vitro, M1 macrophages can still carryout CI (isocitrate-elicited) and CII (succinate-based) respiration even when exposure to NO has reduced protein levels, suggesting that the remaining complexes are operational, and thus inoperative spare complexes have been removed. The inhibition of CI and II of mitochondrial ETC has been thought to result from S-nitrosation of thiols in these protein complexes[49,50]. However, It is known that absence of substrates favors the conversion of CI to its deactivated (D) form, which has exposed cysteines targets of nitrosation and/or oxidation[51,52]. The D form comprises a significant fraction of total CI, and it is revealed in measurements of maximal respiratory capacity (MRC). Indeed, we can rescue MRC with blockade of iNOS during stimulation (Supplementary Fig. 6E, F) and exposure to low dose of NO affects MRC but not basal OCR[53]. Therefore, we propose that, during stimulation, low $NADH/NAD^+$ due to NO-mediated blunting of mentioned enzymatic activities, promotes the accumulation of CI in D form facilitating degradation and favoring full glycolytic commitment. Regardless of the mechanism, oxidation of isocitrate (CI) is maintained indicating that inhibition of aconitase occurs before considerable inhibition of the ETC complexes, in line with previous work[24]. Although we do not know if the disappearance of CII is linked to decline of CI and/or is the result of the same mechanism, we suggest that suppression of OXPHOS during acute NO exposure is due to substrate deprivation rather than via direct effects on mitochondrial ETC. Further experimentation will be required to test this hypothesis.

Given in vivo plasticity and demonstrations that ex vivo manipulation alters cellular redox states and metabolomes[54], the fact that we found that macrophages activated in vivo share many metabolic features with BMDMs in vitro has implications for our understanding of the regulation of immunity and disease pathogenesis. Our data represent one of few investigations of TCA rewiring in vivo. It is worth noting, however, that in our model, peritoneal macrophages did not accumulate citrate as has been demonstrated extensively in vitro. Nevertheless during the modified Schwartzman reaction, peritoneal macrophages changed citrate/isocitrate ratios and exhibited reduced levels of ACO2 activity demonstrating that the "break" in the TCA is a relevant characteristic in vivo. Likely, differences in availability of carbon sources and/or oxygen tension within tissue environments are the reason why in-vivo settings only partially reflect in vitro data sets[35,55].

We are the first to demonstrate alterations in the metabolic signature of the peritoneal lavage during stimulation, where we documented NO-dependent regulation of citrate, α-KG, arginine metabolism and itaconate production. Future studies should determine how peritoneal citrate is controlled by NO and what role that may play during the inflammatory response. Regardless, our study highlights that, during inflammation, the macrophage metabolically responds to the surrounding environment and at the same time, directs alterations of it, presumably to facilitate pathogen clearance. Together with our recent findings[20], our new data suggest that additional work is needed to understand the ramifications of immune cell-mediated metabolic alteration of the peritoneal niche on infection, inflammation and cancer in order to therapeutically manipulate the metabolites within.

The data presented here provide novel interpretation of the biochemical basis of macrophage metabolic programming. This process is orchestrated by NO and is a result and not a direct mediator of polarization. Given the known differences in NO fluxes generated by human and murine macrophages, our data change our understanding of metabolic rewiring and raises the possibility that substantial sources of non-myeloid NO in humans[56] may have metabolic effects by acting in cis or trans on immune cells within the tumor microenvironment.

## Methods

**Mice.** $Nos2^{-/-}$ (B6.129P2-Nos2tm1Lau/J purchased from the Jackson Laboratory) and C57BL/6 control mice were bred and propagated in the same room in the Frederick National Laboratory Core Breeding Facility. Animal care was provided in accordance with the procedures in "A Guide for the Care and Use of Laboratory Animals". Ethical approval for the animal experiments detailed in this manuscript was received from the Institutional Animal Care and Use Committee (Permit Number: 000386) at the NCI-Frederick. All mice used were used between 6 and 10 weeks old and were age- and sex-matched for each experiment.

**Macrophage culture and stimulation.** Total cells from the bone marrow were plated at $7.0 \times 10^5$/mL for 5–7 days in complete Dulbecco's modified Eagles media (DMEM) (Gibco #11965) supplemented with 2 mM Glutamine, 10% FBS, 100 U/mL Penicillin/Streptomycin and 20 ng/mL mouse macrophage colony-stimulating factor (M-CSF) allowing for proper BMDM differentiation and growth[3]. After growing BMDM for 6 days in culture, cells were placed in complete culture media and treated with vehicle (PBS) or 100 ng/mL of LPS or the combination of LPS (100 ng/mL) and 50 ng/mL of IFNγ for 8–24 h. "Activation", "stimulation", and "polarization" in the manuscript are referred to LPS + IFNγ treatment, unless specified. Where indicated, cells where pretreated with aminoguanidine (AG; 1 mM, Sigma Aldrich), dimethyl α-KG (D-αKG; 1 mM, Sigma Aldrich), dimethyloxalylglycine (DMOG; 100 µM, Sigma Aldrich), FeSO₄ (50 µM, Sigma Aldrich)/sodium (Na) thiosulfate (3 mM, Sigma Aldrich), Fluoroacetate (FA; 10–20 mM, Sigma Aldrich) for 1 h as indicated prior to addition of LPS + IFNγ for 8–24 h. Resting cells and LPS/IFNγ-treated $Nos2^{-/-}$ BMDM were additionally treated with DETA/NO (100 up to 500 µM) where indicated.

**RNA/DNA isolation and quantitative PCR.** Cells ($2 \times 10^6$) in wells of 6-well plates were washed and total RNA was extracted using the High-Pure RNA isolation kit (Roche, 11828665001), as per manufacturer's instructions (with the addition that 2-mercaptoethanol (1% v/v) was added to the lysis buffer). cDNA was synthesized from 1 µg RNA using the High-Capacity cDNA Reverse Transcription kit (Applied Biosystems, 4387406) and quantitative RT-PCR was performed with an Applied Biosystems 7300 using Eagle Taq (Roche) reagent and ABI Taqman gene expression probes and normalization was performed against Hprt (Mm01545399_m1). Values are expressed as relative expression using the ΔΔCt method. Data were normalized to WT control.

For measurements of mitochondrial DNA, DNA was extracted with AllPrep DNA/RNA Kit (Qiagen, #80284) and quantitative PCR was carried out with mitochondrial 16 s DNA probe (Mm04260181_s1) and normalized versus β-actin DNA (Mm00607939_s1).

**Immunoblotting.** Protein expression in stimulated macrophages was determined by western blot using 20 µg of total protein per sample[3]. Separation into cytoplasmic and nuclear fractions was performed using 0.4 M NaCl and 0.1% Nonidet P-40 (NP-40) in extraction buffer[57]. SDS immunoblots (IBs) were performed to evaluate steady-state protein levels of structural subunits of mitochondrial complexes assessed by native gels. Antibodies in this study are as follows: anti ACO1/2, anti IDH1, anti SDHB, anti SDHA, anti NDUFS1, anti MTCO1, anti ATP5A, anti NDUFS8, anti total OXPHOS (Complex V, ATP5A subunit; Complex IV, COXII subunit; Complex III, UQCRC2 subunit; Complex II, SDHB; Complex I, NDUFB9 subunit) and anti Phospho-S²⁹³ PDHE1-alpha were from Abcam; anti-FECH antibody was from Proteintech; anti phospho-, total PDK1 and anti HIF1α

were from Novus Biologicals. For loading control anti CS, anti TOM20, Lamin B and anti β-actin were used (respectively from Sigma, Santa Cruz, Abcam, and Millipore).

**Spectrophotometric measurement of aconitase, IDH, and PDH**. Aconitase enzymatic activity was assayed by the disappearance of cis-aconitate at 240 nm measured with a 96-well spectrophotometer plate reader (VersaMax, Molecular Devices): $2 \times 10^6$ stimulated BMDM were lysed with 0.2% Triton X-100 in 0.15 M NaCl buffered with 30 mM Tris-HCl, pH 7.2. Lysate was centrifuged (5,000 g for 15 min) and 100 µg of supernatant was immediately assayed for aconitase activity at 25 °C in the presence of 0.02% BSA. The reaction was started with the addition of 0.2 mM cis-aconitate and enzyme activity was determined from the initial reaction rate. An extinction coefficient of 3.41/cm/ mM was used for cis-aconitate. For mitochondrial aconitase enzymatic activity measurements, $20 \times 10^6$ cells were permeabilized beforehand with 0.007% digitonin to remove cytoplasmic proteins[30].

Isocitrate dehydrogenase was measured spectrophotometrically in both whole-cell and permeabilized-cell lysates by following the reduction of NADP + or NAD + with isocitrate as substrate. Isocitrate (1 mM final concentration) was added to the lysates to start the reaction and the coupled reduction of NADP + (NAD +) (0.1 mM) was followed at 340 nm. Reaction was performed at 37 °C in a buffer containing 30 mM Tris-HCl, 0.15 M NaCl, and 10 mM MgCl₂, pH 7.2[58].

Formation of NADH was used to measure pyruvate dehydrogenase complex activity[59]. The reaction mixture here contained 2.5 mM NAD, 0.1 mM Coenzyme A, 5 mM pyruvate, and 1 mM MgCl₂. A base-line was determined with sample and reference both containing the entire reaction mixture except for pyruvate. Spectrophotometric readings were measured every 20 s for 30 min.

**Metabolic analyses**. OCR and ECAR were examined using the XF96 Seahorse Metabolic Analyzer from Seahorse Biosciences (North Billerica, MA). Briefly, BMDM at day 7 of differentiation were plated at a seeding density of $0.8 \times 10^5$ cells/ well in 200 µL of complete media in an extracellular flux tissue culture plate[35]. BMDM were incubated and stimulated with or without 100 ng/mL of LPS and 50 ng/mL of IFNγ for the times and presence of inhibitors indicated in the figures. Prior to metabolic test the media was removed and replaced with Seahorse XF assay media (#102365-100) containing 25 mM glucose, 2 mM glutamine, and 20 ng/mL M-CSF and the plates containing cells were incubated for 30 min at 37 °C with no CO₂. Port additions and times were used as indicated in the figures.

For mitochondrial stress test (extracellular flux analysis), 1.26 µM oligomycin was injected in port A, 0.67 µM FCCP fluoro-carbonyl cyanide phenylhydrazone in port B, and 0.2 µM rotenone/1 µM antimycin A in port C. For experiments of longer OCR monitoring in BMDMs throughout addition of stimulants, the LPS + IFNγ mixture was injected in port A and after 9 h, dichloroacetate (DCA; 10 mM, Sigma Aldrich) or FA (10 mM) were injected in port B. For glutamine deprivation studies, prior to metabolic test the media was replaced with complete Seahorse XF media without glutamine.

For respiration measurements, OCR was examined using the XF96 Seahorse Metabolic Analyzer[60]. Briefly, BMDM at day 7 of differentiation were plated at a seeding density of $0.8 \times 10^5$ cells/well in 200 µL of complete media and activated O. N. Upon XF96 run, media was replaced with MAS buffer (220 mM Mannitol, 70 mM sucrose, 10 mM KH₂PO₄, 5 mM MgCl₂, 2 mM HEPES, 1 mM EGTA, 0.2% fatty acid free BSA). Oxygen consumption was monitored permeabilizing cells with 1 nM PMP (Seahorse Biosciences). State III respiration, defined as ADP-stimulated respiration in intact, unpoisoned mitochondria, in the presence of excess substrate was assessed[61]. Citrate (5 mM), ADP (4 mM) and the mentioned permeabilizer were co-injected in port A. Isocitrate (5 mM) was injected in port B; 5 mM tartronate (2-hydroxy malonate) was also added to the respiration medium as an exchange partner for the tricarboxylate-transporter, which facilitates the entry of citrate and isocitrate into the mitochondrial matrix[21,30]. Rotenone 0.2 µM was injected in port C.

Pyruvate-elicited respiration was assessed in permeabilized cells offered 5 mM pyruvate, 2.5 mM malate (port A), and 0.2 µM rotenone (port B)[60]. For complex II-mediated respiratory activity 10 mM succinate, in the presence of 0.2 µM rotenone, was added upon permeabilization. 1 µM antimycin A was injected in port B. The concentration of the above mentioned substrates are final concentrations.

Glucose uptake was assayed in cultured BMDM using a hand-held glucometer (Accu-Check, Roche, Diagnostics, IN) or measuring its levels with GC-MS where a control sample of complete DMEM incubated at the same experimental conditions without cells was provided.

**Metabolomics and ¹³C tracing studies**. WT and $Nos2^{-/-}$ BMDM (~$10^7$ cells per sample) were treated with vehicle (PBS) or 100 ng/mL of LPS for 24 hours. Cells were then washed, gently scraped, pelleted, snap frozen in liquid nitrogen and sent to West Coast Metabolomic Center at the University of California, Davis (Davis, CA) for metabolomic analyses. Same pipeline was applied for WT and $Hif1\alpha^{-/-}$ BMDM treated O.N. with vehicle (PBS) or LPS + IFNγ for untargeted studies.

Targeted measurements were performed through electrospray ionisation mass spectrometry (ESI-LC–MS/MS) analysis in multiple reaction monitoring mode with an Agilent 6410B Triple Quadrupole mass spectrometer interfaced with a 1200 Series HPLC quaternary pump (Agilent) available in house at NCI-Frederick.

Cell pellets were washed and resuspended in 80% methanol and 50 µL or 4x concentrated mouse peritoneal lavage were mixed with 450 µL of 100% methanol. Phase separation was achieved by centrifugation at 4 °C and the methanol-water phase containing polar metabolites was separated and dried using a vacuum concentrator. The dried metabolite samples were stored at −80 °C and resuspended in milli-Q water the day of analysis.

Four concentrations of standards, processed under the same conditions as the samples, were used to establish calibration curves[62]. The best fit was determined using regression analysis of the peak analyte area. Chromatographic resolution was obtained in reverse phase on a Zorbax SB-C18 (1.8 µm; Agilent) for aminoacids and a Eclipse Plus C18 (1.8 µm; Agilent) for TCA intermediates, with a flow rate set at 0.4 mL/min.

For ¹³C-carbon incorporation from [U-¹³C] glucose or glutamine in metabolites, after O.N. exposure to LPS/IFNγ (day 7 of differentiation), cells were washed with PBS and DMEM containing dialyzed FBS and labeled substrate, i.e., [U-¹³C] glucose or [U-¹³C] glutamine (Cambridge Isotope Laboratories) was added for 4 h (confirmation of steady-state). For mass spectrometry analysis, cells were scraped in 80% methanol and phase separation was achieved by centrifugation at 4 °C and the methanol-water phase containing polar metabolites was separated and dried using a vacuum concentrator. The dried metabolite samples were stored at −80 °C. Isotopologue distributions and metabolite levels were measured with a 7890A GC system (Agilent Technologies) combined with a Quattro Premier MS system (Waters)[63]. $m + 0$ to $m + n$ indicate the different mass isotopologues for a given metabolite with n carbons, where mass increases due to ¹³C-labeling[64].

For NMR analysis, labeled cells were rinsed in cold PBS, quenched in cold acetonitrile, extracted for metabolites, and prepared in a solution of 1xDSS-PO₄ 50% D₂O. NMR spectra were recorded at 14.1 T under standard acquisition conditions using 1D proton and 1D [1H]{[13C]}-heteronuclear single-quantum coherence (HSQC) for isotopomer analysis. The HSQC analysis compared the peak intensity of protons attached to [13C] atoms (akin to [13C] abundance) at specific positions of various metabolites. Peaks were assigned and quantified[65]. The 1-D HSQC spectra were normalized to protein content and spectral parameters so that intensity or metabolite resonances reflected their protein amount.

**Isolation of mitochondria and immunoprecipitation**. Briefly, mitochondria from BMDM pellets (~$10^7$ cells) were isolated from the cytosolic fractions after cell permeabilization with a buffer containing 0.1% digitonin in 210 mM mannitol, 20 mM sucrose, and 4 mM HEPES. The pellets after centrifugation at 700 x g for 5 min contained mitochondria, which were isolated by differential centrifugation and solubilized in lysis buffer I containing 50 mM Bis-Tris, 50 mM NaCl, 10% w/v Glycerol, 0.001% Ponceau S, 1% Lauryl maltoside, pH 7.2, protease and phosphatase inhibitors[66].

Isolated mitochondria were lysed in a buffer containing 25 mM Tris, 200 mM NaCl, 1 mM EDTA, 1% NP-40, 5% glycerol (pH 7.4) to obtain a final protein concentration of 1 µg/µL. Five-hundred micrograms of total mitochondrial proteins were used for the IP with the co-IP kit from Pierce (#26149). Beads with covalently bound antibody (2 µg) (anti-IgGs was used as a negative control) were incubated with mitochondrial lysate ON at 4 °C. After, beads were washed five times with lysis buffer, and the bound complexes were finally eluted with Tris/Gly pH 2.8 (50 µL/ IP sample) at room temperature for 10 min.

**Native PAGE (BN-PAGE) and native immunoblots**. The Native PAGE Novex Bis-Tris gel system (Thermo Fisher Scientific) was used for the analysis of native membrane protein complexes and native mitochondrial matrix complexes; 20 µg of membrane protein extracts were loaded/well; the electrophoresis was performed at 150 V for 1 h and 250 V for 3.5 h[66]. For the native IBs, polyvinylidene fluoride was used as the blotting membrane. The transfer was performed at 25 V for 4 h at 4 ºC. After transfer, the membrane was washed with 8% acetic acid for 20 min to fix the proteins, and then rinsed with water before air-drying. The dried membrane was washed 5- 6 times with methanol (to remove residual Coomassie Blue G-250), rinsed with water and then blocked for 2 h at room temperature in 5% milk, before incubating with the desired antibodies diluted in 2.5% milk O.N. at 4 ºC. In order to avoid strip and reprobing of the same membrane, which might allow detection of signals from the previous IBs, samples were loaded and run in replicates on adjacent wells of the same gel, and probed independently with different antibodies.

**Complex I, complex II, and complex IV in-gel activities**. For complex I activity, after resolution of the respiratory chain complexes by BN-PAGE, the gel was incubated with 0.1 M TrisCl, pH 7.4, containing 1 mg/mL nitrobluetetrazolium chloride (NBT) and 0.14 mM NADH at room temperature for 30–60 min. For complex II, detection of succinate CoQ-reductase activity (SQR) (CoQ-mediated NBT reduction) was performed by incubating the gel for 30 min with 84 mM succinate, 2 mg/mL NBT, 4.5 mM EDTA, 10 mM KCN, 1 mM sodium azide and 10 µM ubiquinone in 50 mM PBS, pH 7.4. For complex IV, the gel was incubated in 50 mM phosphate buffer pH 7.4 containing 1 mg/mL DAB (3,3'-diaminobenzi-dine) and 1 mg/mL cytochrome c at room temperature for 30-45 min[66].

**PDHE3 in-gel activity**. 30 μg of mitochondrial matrix lysates were run on Blue Native PAGE at 150 V for 1 h, followed by 1 h at 250 V on ice. The gel was soaked into 50 mM potassium phosphate buffer (pH 7.0) assay buffer, containing 0.2 mg/mL NBT and 0.1 mg/mL NADH for 30–45 min until DLD activity developed[47].

**Microarray**. WT and $Nos2^{-/-}$ BMDM ($1 \times 10^6$ cells/mL in 10 cm Petri dishes) were treated with vehicle (PBS) or 100 ng/mL of LPS and 50 ng/mL of IFNγ for 8 h. Cells were then washed and total RNA was extracted using the High-Pure RNA isolation kit (Roche, 11828665001), as per manufacturer's instructions (with the addition that 2-mercaptoethanol (1% v/v) was added to the lysis buffer). One microgram of RNA from each sample was then sent to CCR—Genomics, Microarray Core Service, LMT Gene Expression Group at the Frederick National Laboratory for Cancer Research, Frederick (Frederick, MD) for Affymetrix Mouse Genome 430 2.0 Array.

For differential expression analysis, we input CEL files into Partek Genomic Suite software (Partek Inc.) and sample treatment information was manually annotated. Data were processed by the build-in gene expression pipeline, which included the RMA algorithm for data normalization and converted to log2 values. Data were deposited in GEO NCBI under the accession GSE115120. Differentially expressed genes were identified by analysis of variance (ANOVA). The threshold value has been set at 1.5-fold change and a false discovery rate (FDR) < 0.5. Differential expressed genes were analyzed by ingenuity pathway analysis (IPA; Ingenuity Systems).

**Cytokine production**. For cytokine analysis BMDM at day 7 of differentiation were plated at a seeding density of $1 \times 10^5$ cells/well in 200 μL of complete media in wells of 96-well plates. Plates were incubated and stimulated with or without 100 ng/mL of LPS and 50 ng/mL of IFNγ in the absence or presence of inhibitors (where indicated in the figures) before stimulation. The plates were incubated for 24 h before supernatant removal. The cytometric bead array kits (BD Biosciences) were used to quantify IL10, IL1β, Mip1α and KC in 50 μL supernatant, while a 1:5 dilution was used for detection of IL6, IL12p40, TNFα, and MCP1 cytokines. Kits were used as per manufacturer's instructions.

**Shwartzman reaction model of sepsis in mice**. Macrophages were recruited into the peritoneum by an i.p. injection of 0.5 mL of 4% Brewer's thioglycollate broth (Difco, Detroit, MI)[67,68]. Mice were placed on a heating pad and monitored daily. Seventy-two hours later sepsis was induced by the Shwartzman reaction with the priming injection of 15 μg IFNγ i.p. followed 12 h later by a challenge LPS injection[39,69]. The optimal dose of LPS ranged between 30 and 100 μg (1.25–4 mg/kg). Cells were obtained lavaging mouse peritoneum 12 h post challenge with 4 mL PBS using a 21-G needle.

Cell purification from peritoneal lavage fluid was performed by magnetic purification. Neutrophil depletion was achieved incubating total cells with α-Ly-6G-Biotin in the presence of PBS with 5 mM ethylenediaminetetraacetic acid (EDTA), 0.5% BSA for 15 min at 4 °C and sorting with streptavidin microbeads and LS columns as per manufacture's instructions (Miltenyi Biotec). Macrophage were isolated from flow throughs, carried out with addition of CD11b microbeads with incubation for 20 min at 4 °C followed by positive selection with magnetic columns (Miltenyi Biotec).

Cell pellets were washed with NaCl 154 mM and subjected to metabolite extraction for targeted metabolomics studies, together with concentrated lavage fluid.

**NO detection**. Nitrite, the oxidation product of NO, was measured using the Griess reaction according to the manufacturer's instructions (Invitrogen™, #G7921).

**Statistical analysis**. Data are presented as means ± standard error of mean (± SEM) and are representative of at least two independent experiments. Statistical analysis was performed using GraphPad Prism 7. One-way or two-way ANOVAs followed post tests and linear regression were used when multiple groups were analyzed as are indicated in figure legends. Student's $t$-test was used for statistical analysis when two groups were analyzed. All data are assumed normally distributed and were log transformed where appropriate before analysis. Statistical significance was set at $*p < 0.05$.

**Reporting summary**. Further information on research design is available in the Nature Research Reporting Summary linked to this article.

## Data availability

All data supporting the findings of this study are available with the article, and can also be obtained from the corresponding author. Microarray data has been deposited in the NCBI GEO repository with an accession ID GSE115120. Metabolomics data is available at the NIH Common Fund's National Metabolomics Data Repository (NMDR) website,

the Metabolomics Workbench, where it has been assigned Project ID PR000867. The data can be accessed directly via it's Project https://doi.org/10.21228/M8MM6Z. The source data underlying Figs. 1–5 and 7 and Supplementary Figs. 1–7 are provided as a Source Data file.

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

## Acknowledgements

We thank Prof. Bernhard Brüne at Goethe-University Frankfurt- Faculty of Medicine for providing *Hifα1−/−* BMDMs, Jeffrey Subleski (Leukocyte Signaling Section, Cancer & Inflammation Program, National Cancer Institute, Frederick, MD, USA) for its technical assistance and Daniel R. Crooks (Urologic Oncology Branch, National Cancer Institute, Bethesda, MD, USA) for helpful discussions. This research was supported, in part, by the intramural Research Program of the NIH, National Cancer Institute USA, 1U24DK097215-01A1 (to RMH, TWMF, and ANL), and Redox Metabolism Shared Resource(s) of the University of Kentucky Markey Cancer Center (P30CA177558). L.C.D. is funded in part by and the Henry Wellcome Trust, UK (WT103973MA). The content of this publication does not reflect views or policies of the Department of Health and Human Services, nor does mention of trade names, commercial products, or organization simply endorsement by the U.S.Government.

## Author contributions

E.M.P. conceived and performed experiments, and wrote the manuscript. M.G.C, W.A.B. and N.M. performed experiments. D.W.M helped design studies, wrote the manuscript and secured funding. L.C.D, M.G.C. and C.M.R. conducted formal analysis and critically improved the manuscript. B.G., T.C, T.W.M.F, R.M.H. and A.N.L. acquired and analyzed the tracer data. T.W.M.F. and A.N.L. contributed to tracer data interpretation. D.A.W and T.A.R. provided reagents, expertize, and feedback.

## Competing interests

The authors declare no competing interests.
