## [Peer Review File · Nature Communications]

Reviewers' comments:

Reviewer #1 (Remarks to the Author):

In this manuscript, Erika M. Palmeiri et al. aim to determine the role of NO in orchestrating the metabolic reprogramming and the TCA cycle alterations associated with M1 macrophage polarization. By comparing WT macrophages with NO₂^{-/-} macrophages, they propose that nitric oxide is responsible for those metabolic changes including glycolytic activity and TCA alterations. And they investigate the underlying mechanism. Their results exclude the involvement of the Isocitrate Dehydrogenase 1 whose downregulation is currently believed to be the main cause of TCA cycle break in M1 macrophages. They provide evidence which indicates that NO accumulation reshapes the metabolic landscape of M1 macrophage by inhibiting the mitochondrial aconitase and the pyruvate dehydrogenase but promotes the glutamine utilization. Finally, they show that the NO accumulation lead to the loss of some of the mitochondrial electron transport chain complexes. Although this NO-mediated metabolic reprogramming in inflammatory macrophages has already been described, the mechanisms proposed in this manuscript are interesting. However, the connections are often based on correlation, not on direct relations and the paper is too descriptive. Moreover, the interpretation is not fully supported by the results and several references are inappropriately cited.

1. The link between the NO and the inhibition of ACO₂ is never really established. The rescue experiment performed with the ferrous iron is fine but fail to prove the impact of ACO₂ inhibition on the TCA cycle. Some further experiments are needed to confirm this causal relation.
2. For the last experiment the link between NO and the inhibition of some of the ETC complexes is not well defined either: the results don't allow to conclude that the ETC functional impairment is NO-dependent. It doesn't really add anything valuable to the article. For example, some rescue experiment would be appropriate to prove concretely the link or to propose some mechanism controlling this impairment.
3. The metabolic analyses shown in figure 1 is problematic. First, despite the authors claim that Nos2-ko macrophages have lower ability to convert glucose into lactic acid, the result in Figure 1b still show increase of lactic acids. How can Nos2-KO macrophages generate such high levels of lactic acids (comparable to wt macrophage, as supported by supplementary figure 1b) if the mechanisms claimed by the authors are operating? In addition, Figure 1b suggests that Nos2-ko macrophages remain effectively engage glycolysis. Although the authors calculate glycolytic rates in supplementary figure 1d, this method is not commonly used. Seahorse analysis should be done to properly address this.
4. In Figure 1c, the authors showed that Nos2-KO macrophages have lower aconitase levels, but a bit higher itaconate. Although this fit with what they propose on ACO₂ regulation, IRG1 inhibitor should be used to examine whether Nos2-KO macrophages can accumulate more aconitase. This can further strengthen the conclusion of this finding.
5. It is not clear how Figure 1H was done and analyzed. Is there any repeat and statistic analysis?
6. The authors claim that reprogramming of the mitochondrial metabolism should be considered a result of inflammatory polarization rather than a mediator of this process. Although in figure 1, they showed that M1 activation was not affected in the absence of mitochondrial metabolism change. The results presented in this manuscript do not allow the authors to make this conclusion since the key process of lactic acid production remains, the author should tone down their original claim. Moreover, whether M2 marker genes will be affected should also be examined.
7. For how NO regulate PDH enzyme activity, S-Nitrosation, a mechanism of posttranslational protein modification mediated by NO, has been shown to regulate protein activity, is there any possibility that NO-mediated PDH activity is through this form of posttranslational modification.

8. It would be great if the authors could validate this mechanism in animal studies.

9. In the first paragraph of introduction, the authors described shifts toward Warburg metabolism in macrophages has been characterized. However, in this sentence they cite two references about dendritic cells! This also happened again in the last sentence of this paragraph. Some of citation errors also happen in the remaining part of this manuscript, especially roiginal research works were not cited instead the authors cited review article to support their claim.

Minor corrections:

- Only two samples for each condition on Fig2 D?
- Protein expression of glutaminase in Fig5 B should be examined.

Reviewer #2 (Remarks to the Author):

Palmieri and colleagues investigated the metabolic reprogramming during LPS+IFNg-induced classical macrophage polarization. They identified the important role of NO in the alteration of TCA cycle and mitochondrial respiration in classically polarized mouse BMDMs. The experiments are high quality and the manuscript is well written. The plethora of the rigorous metabolomic studies are particularly valuable, but these are entirely descriptive and provide very little mechanistic insights. Also in vivo pieces of evidence documenting that the NO-induced metabolic changes have functional consequences in inflammatory macrophages are completely missing.

Major points of criticism:

1. It would be important to show some experimental data that NO can also regulate the macrophage metabolism including Aconitase 2 activity, levels of TCA cycle intermediates and mitochondrial Electron Transport Chain complexes in inflammatory conditions in vivo. The usage of very complex in vivo inflammatory models is not necessary, the study of peritoneal macrophages following intraperitoneal LPS injection is sufficient.
2. The authors demonstrated that NO accumulation leads to the loss of mitochondrial Electron Transport Chain complexes in classically polarized macrophages. However, it would be informative to demonstrate whether NO is able to modulate the number of mitochondria during classical macrophage polarization.

Response to Reviewer #1

Reviewer #1 (Remarks to the Author):

The new version of the manuscript shows some improvements from the mechanistic point of view. Indeed, the authors could prove that the PDH activity is directly compromised by NO and that the ETC downregulation during M1 polarization is NO-dependent as well.

The authors provide some in vivo evidences that what they observe in vitro occurs also in peritoneal macrophages upon IP injection of IFN and LPS.

We thank the reviewer for the positive comments. We agree that the new experiments performed substantially improved the manuscript adding mechanisms to NO-mediated restriction of PDH and regulation of the function of the mitochondrial ETC complexes. Additionally we believe that the added in vivo data strongly validate this mechanism and that our study of peritoneal macrophages reveals the compelling capacity of peritoneal microenvironment to be altered metabolically by NO.

However, the general feeling is that the paper misses some insight on how, by interfering with the TCA cycle and oxidative respiration, NO affects the M1 polarization. Is there modification in the transcriptional activation and molecular signalling? Why is the M1 signature increased in NOS2 KO cells? Without these information, this manuscript is relatively descriptive on metabolic actions controlled by NO, but falls short on providing its regulatory circuits on macrophage behaviour.

Here the reviewer hits on an enormous point. We were surprised to find that the profound, NO-dependent, metabolic rewiring associated with M1 polarization is not directly required for adoption of the M1 inflammatory phenotype. Because there continues to be misunderstanding in the field regarding the basis, mechanisms, and targets of the rewiring, the act of rewiring has incorrectly been assumed to be part of the development of the inflammatory phenotype. Here we intend to emphasize that the NO-dependent remodeling we describe is the result of polarization, not a mediator of polarization (see lines 638-640). Now that we have revealed the mechanisms and targets of this activity, the field can now begin to dissect the downstream ramifications of the presence or absence of these specific events.

A clear possibility for downstream effects of reprogramming is alterations of transcriptional profiles. Indeed, we demonstrate here a substantial transcriptional impact of NO during stimulation (Fig. 1H and Table 1 and S1). In future work, we hope to dissect the degree to which these events are tied to NO control of specific events involved in rewiring including alterations in itaconate, acetyl-CoA, α -ketoglutarate, and/or S-adenosylmethionine (SAM) metabolism. Alternatively, many of these events may be a direct result of NO through redox-related modifications. The increase in the M1

inflammatory program of *Nos2*^{-/-} cells the reviewer refers to is an example of the likely complexity. As we discuss in the manuscript, the enhanced expression of inflammatory genes may be due to impaired Nrf2 activation in the absence of NO as Nrf2 is known to attenuate inflammation in myeloid cells. However, given the recent data suggesting Nrf2 activation by itaconate, a metabolite increased in *Nos2*^{-/-} cells, this question now requires substantial additional dissection to fully understand the interplay. This is a substantial area of research as it impacts how we really think about the immunometabolism associated with macrophage polarization.

A “more functional” in vivo readout would have probably helped to understand whether this whole mechanism is really relevant for pro-inflammatory macrophages to act in that way. Although the authors used peritoneal macrophage activation to show what they observed can be elicited in vivo, this is very different from its impact on disease condition.

Although our in vivo system is not a true disease model, we feel that our demonstration of the in vivo dependence of many metabolic events on NO provide exciting and novel potential explanations for this discrepancy. Given the limitations of cell numbers needed for metabolic profiling and complexity of in vivo experiments, we followed the advice of the other assigned reviewer and developed a model based on endotoxic shock that used well characterized ligands in a tightly controlled system. We felt this approach was most likely to yield a fairly homogenous population of M1 macrophages with the numbers and reproducibility that would permit our detailed analysis.

It is worth noting again, that currently there are very few examples indicating that macrophage populations undergo even general metabolic rewiring in vivo, and even rarer that detail specific mitochondrial effects and/or products. Despite its limitations, the model reviewer #2 suggested did provide important insight. The in vivo metabolic characterization of peritoneal cells and fluid we describe here, revealed a metabolic perturbation of the microenvironment caused by macrophages and their production of detectable nitric oxide levels. These interesting findings are “functional” in the sense that they contribute to the understanding of macrophage phenotypes and to how we can decipher metabolite niches during inflammation with an eye to eventually therapeutically manipulating energy metabolism, as well as citrate and itaconate production, in physiological and pathological conditions.

REVIEWERS' COMMENTS:

Reviewer #1 (Remarks to the Author):

The responses of the authors are reasonable to me. Based on the current information in metabolic regulation in macrophages and the information provided here, this study will provide more information for better define this field. Based on the value and the quality of this work, it should be published.

Response to Reviewer #1

REVIEWERS' COMMENTS:

Reviewer #1 (Remarks to the Author):

The responses of the authors are reasonable to me. Based on the current information in metabolic regulation in macrophages and the information provided here, this study will provide more information for better define this field. Based on the value and the quality of this work, it should be published.

We thank the reviewer for the positive comments.